# Integrating Deep Metric Learning with Coreset for Active Learning in 3D Segmentation

**Arvind Murari Vepa**
UCLA
amvepa@ucla.edu

**Zukang Yang**
UC Berkeley
zukangy@berkeley.edu

**Andrew Choi**
Horizon Robotics
asjchoi@ucla.edu

**Jungseock Joo**
UCLA
jjoo@comm.ucla.edu

**Fabien Scalzo**
UCLA
fab@cs.ucla.edu

**Yizhou Sun**
UCLA
yzsun@cs.ucla.edu

## Abstract

Deep learning has seen remarkable advancements in machine learning, yet it often demands extensive annotated data. Tasks like 3D semantic segmentation impose a substantial annotation burden, especially in domains like medicine, where expert annotations drive up the cost. Active learning (AL) holds great potential to alleviate this annotation burden in 3D medical segmentation. The majority of existing AL methods, however, are not tailored to the medical domain. While weakly-supervised methods have been explored to reduce annotation burden, the fusion of AL with weak supervision remains unexplored, despite its potential to significantly reduce annotation costs. Additionally, there is little focus on slice-based AL for 3D segmentation, which can also significantly reduce costs in comparison to conventional volume-based AL. This paper introduces a novel metric learning method for Coreset to perform slice-based active learning in 3D medical segmentation. By merging contrastive learning with inherent data groupings in medical imaging, we learn a metric that emphasizes the relevant differences in samples for training 3D medical segmentation models. We perform comprehensive evaluations using both weak and full annotations across four datasets (medical and non-medical). Our findings demonstrate that our approach surpasses existing active learning techniques on both weak and full annotations and obtains superior performance with low-annotation budgets which is crucial in medical imaging. Source code for this project is available in the supplementary materials and on GitHub: https://github.com/arvindmvepa/al-seg.

## 1 Introduction

In the field of 3D medical segmentation, manual annotation of entire volumes, despite being laborious and time-consuming, has been the gold standard. Annotating a single 2D image can take minutes to hours depending on the complexity of the image (75; 73; 15; 71; 6), and a 3D medical volume, containing up to 200 slices, can require a significant amount of expert labor. Annotating a full dataset not only imposes a significant time burden on medical experts but also incurs high costs. Therefore, active learning (AL) is urgently needed to optimize annotation efforts.

Surprisingly, the potential of AL in the context of 3D medical segmentation has not been extensively explored. Traditional AL techniques typically focus on either diversity or model uncertainty, often neglecting relevant groupings within the data. For example, slices from the same patient or volume tend to show consistent characteristics. We propose a deep metric learning strategy that identifies and utilizes these similarities to better highlight diversity in the active learning process. Diverse samples

38th Conference on Neural Information Processing Systems (NeurIPS 2024).

help the model learn a wide variety of patterns and features, which can be crucial for generalization. While medical imaging has natural groupings which we can leverage, our approach extends to a wide range of real-world datasets, including video segmentation.

Several other strategies may also help in reducing costs. Most current AL approaches use volume-based AL (39) rather than slice-based AL for 3D segmentation, which tends to be more costly and less efficient. Alternatively, weakly supervised methods, which require simpler annotations like scribbles (34; 29; 11), bounding boxes (13; 83), points (82), or semi-automated techniques (5; 44; 62; 56), have been shown to perform comparably to fully supervised methods (72; 40; 54; 13; 26; 59). However, combining AL with weak supervision, especially in medical settings, remains unexplored. In our research, we explore both slice-based AL and the integration of AL with weak supervision as potential cost-reducing measures.

In this paper, we present our contributions to AL for 3D medical segmentation:

1. The first work to integrate deep metric learning with Coreset during active learning for 3D medical segmentation. Our approach shows superior performance across four datasets (medical and non-medical) with low annotation budgets.
2. The first work to comprehensively compare new and existing algorithms for slice-based active learning for 3D medical segmentation utilizing both weak and full annotations.

## 2   Related work

**Active learning**    AL methods can be broadly classified into 1) uncertainty-based and 2) diversity-based methods (76). Uncertainty-based methods include deep bayesian methods (21; 45; 38; 65), deep ensembles (3; 14; 52; 32; 20), contrastive learning methods (85; 41; 87; 36), and geometry-based methods (19). Diversity-based methods include coreset-based methods (64; 7), clustering-based methods (23), variational adversarial learning methods (66; 37), and random sampling methods (48). A limitation of previous methods is that they are too general and fail to utilize common data groupings found in real-world datasets. Prior methods (67; 9; 69) have tried to incorporate groupings but do not utilize domain information to generate them, which can be suboptimal. Recent research has started integrating domain-specific data groupings into AL algorithms with some success(28; 79), but these methods are designed for specific uses and lack broad applicability. While other methods have adapted Coreset (31; 30), they are not specifically tailored to 3D medical segmentation as our method is.

**Deep metric learning**    Metric learning is focused on developing methods to measure similarity between data points and used in many applications. Recently, metric learning has focused on deep learning-based feature representations for data points (46). Contrastive losses are popular for metric learning, including Triplet Loss (63) and NT-Xent loss (67). Several non-contrastive approaches have been proposed based on center loss (77; 17; 16), proxy-based methods (47; 70; 27), and LLM guidance (61). One interesting approach is ensemble deep metric learning which combines embeddings from an ensemble of encoders (1; 43; 53; 60; 80). Recent work has improved on ensemble-based methods by factorizing the network training based on differerent objectives (74). However, these approaches can be computationally expensive and narrowly tailored to specific applications. Additionally, non-contrastive approaches often require class supervision to perform well. While some non-contrastive approaches do not (88; 18), they are also narrowly tailored to specific applications.

In contrast, contrastive learning is often used in self-supervised settings in diverse applications (10; 2; 55). The NT-Xent loss specifically outperformed other contrastive losses in self-supervised zero-shot image classification and outperformed fully-supervised ResNet-50 (12). However, prior work (including in active learning (85; 23; 39)) does not leverage any inherent data groupings in the contrastive loss, which can be useful weak supervision. Additionally, recent work in active learning can be computationally expensive because they retrain the feature encoder after each active learning round (85).

**Active learning in medical segmentation**    There has been some notable research on AL in medical segmentation. Earlier work utilized bootstrapping to estimate sample uncertainty (81). In another work, the researchers built a mutual information-based metric between the labeled and unlabeled pools to improve diversity (49). However, both of these approaches are computationally expensive

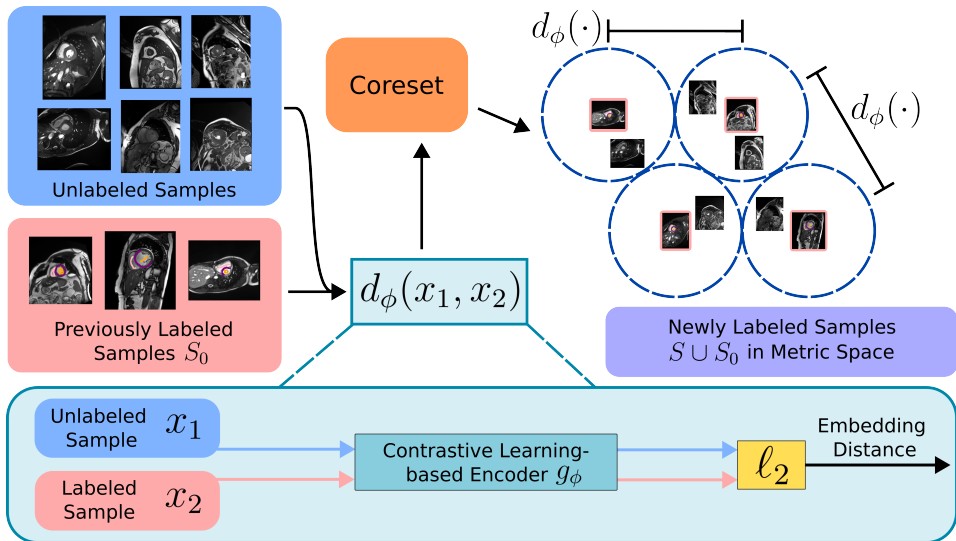

Figure 1: Overview of our active learning pipeline

and not scalable for large datasets. There were also inefficiencies in previous studies, such as choosing whole volumes instead of individual slices in 3D segmentation, which increased costs (39). Random sampling proved effective in some cases (6) and calculating uncertainty using stochastic batches was also effective (20). None of the prior approaches leverage the data groupings inherent in medical 3D data and also do not focus on active learning with weak annotations (e.g., scribbles). While another method also utilizes groupings in medical data (86), their groupings are assumed to be quite large and it's unclear how they would extend their group classification approach to when there are a large number of patients, volumes, and adjacent slice groups like with our datasets.

## 3 Methodology

### 3.1 Problem Formulation

In this section, we formally describe the problem of active learning for 3D segmentation. Let $\mathcal{X} \subset \mathbb{R}^{h \times w \times d}$ be the set of 3D volumes and $\mathcal{Y} \subset \{0,1\}^{h \times w \times d \times k}$ be the set of 3D masks where $h, w, d$ correspond to the height, width, and depth (number of slices) of a 3D volume and $k$ refers to the number of classes. In 3D segmentation, we learn a mapping $F : \mathcal{X} \to \mathcal{Y}$.

Consider a loss function $\mathcal{L} : \mathcal{F} \times \mathcal{Y} \to \mathbb{R}$ where $\mathcal{F}$ is the range of model prediction probabilities ($\mathcal{F} = [0,1]^{h \times w \times d \times k}$). We consider the dataset $D$ a large collection of data points which are sampled *i.i.d.* over the space $\mathcal{Z} = \mathcal{X} \times \mathcal{Y}$ as $\{\mathbf{x}_i, \mathbf{y}_i\}_{i \in [n]} \sim p_{\mathcal{Z}}$. We additionally consider a partially labeled subset $s \subset D$. Thus, active learning for 3D segmentation can be formulated as follows:

$$\underset{\Delta s \subset D, |\Delta s| \leq k}{\operatorname{argmin}} E_{\mathbf{x}, \mathbf{y} \sim p_{\mathcal{Z}}}[\mathcal{L}(\hat{\mathbf{y}}, \mathbf{y})] \tag{1}$$

where $\hat{\mathbf{y}}$ are the model predictions $F(V)$ (where $V = [\text{slice}_1, ..., \text{slice}_d]$ is the volume), $\Delta s$ is the optimal requested labeled set, $k$ is the active learning budget, and $F$ is the deep learning method learned from $s \cup \Delta s$. The Coreset approach is one method of solving Equation 1 (64). However, there are two distinguishing factors in our current work: 1) slice-based active learning for 3D segmentation and 2) deep metric learning. In slice-based 3D segmentation, we learn a mapping $f$ from $\mathcal{X}' \to \mathcal{Y}'$ where $\mathcal{X}' \subset \mathbb{R}^{h \times w}$, $\mathcal{Y}' \subset \{0,1\}^{h \times w \times k}$, and $F(V) = [f(\text{slice}_1), ..., f(\text{slice}_d)]$. Additionally, $D$ is a collection of slices which are sampled *i.i.d.* over the space $\mathcal{Z}' = \mathcal{X}' \times \mathcal{Y}'$ as $\{\mathbf{x}'_i, \mathbf{y}'_i\}_{i \in [n]} \sim p_{\mathcal{Z}'}$.

In the original Coreset paper (64), $s \cup \Delta s$ is the set cover of $D$ with radius $\delta$. However, the Euclidean metric used to calculate the radius does not utilize task-relevant information and may be sub-optimal. This motivates us to consider deep metric learning to utilize more task-relevant information, where $d_{\phi}(x_1, x_2)$ is some parameterized metric. In the the original paper (64), the authors provide a theorem

which bounds the loss from Equation 1 based on the radius $\delta$. We show a very similar bound where $\delta = \max_{x_1 \in D} \min_{x_2 \in s \cup \Delta s} d_\phi(x_1, x_2)$ (we defer the precise bound and proof to the Appendix B). This leads to our Coreset optimization formulation:

$$\underset{\Delta s \subset D, |\Delta s| \leq k}{\operatorname{argmin}} \max_{x_1 \in D} \min_{x_2 \in s \cup \Delta s} d_\phi(x_1, x_2) \tag{2}$$

The above formulation can be intuitively defined as follows; choose $k$ additional center points such that the largest distance between a data point and its nearest center is minimized (64).

## 3.2 Metric learning

The parameterized metric $d_\phi(x_1, x_2)$ is defined as:

$$d_\phi(x_1, x_2) = \ell_2(g_\phi(x_1), g_\phi(x_2)) \tag{3}$$

where $\ell_2$ is the Euclidean metric. Our goal is to learn $g_\phi$, or a feature representation, that emphasizes task-relevant similarities and differences in the data for selecting diverse samples for Coreset. We do this by training a contrastive learning-based encoder with a unique Group-based Contrastive Learning (GCL) which utilizes inherent data groupings specific to medical imaging to generate the feature representation. $d_\phi$ is then used by the Coreset algorithm to select the optimal set of slices. A flow chart illustrating our pipeline can be seen in Figure 1. The annotations for the collected slices are then used to train a segmentation model.

### 3.2.1 Group-based Contrastive Learning for feature representation in metric learning

While there may be several task-relevant groupings in 3D medical imaging, it is not immediately apparent which of these would be useful for feature representation for metric learning for Coreset. For the ACDC dataset, we note the mean pairwise absolute deviation of the normalized training slice pixel values within different groups averaged over the dataset: 0.217 over the entire dataset, 0.159 within patient groups, 0.166 within volume groups, and 0.115 within adjacent slice groups. Note that the volume groups and the adjacent slice groups are the most and least diverse respectively. While intuitively the most diverse group would have the most important features for diversity, this does not necessarily indicate what combinations of groups would be helpful as well.

Instead, we propose a general group contrastive loss based on NT-Xent loss (67). The NT-Xent loss focuses on generating and comparing embeddings for image pairs and their augmentations. It promotes similarity in embeddings for the same image and its augmentation while encouraging dissimilarity for different images. In the same vein, in our group-based loss, we promote similar embeddings for slices from the same group and dissimilar embeddings for slices from different groups, enhancing group-level representation. We define "group" as a set of 2D slices associated with one patient. The formula is as follows:

$$\mathcal{L}_{\text{group}} = -\frac{1}{NG} \sum_{i=1}^{N} \sum_{j \neq i, g_j = g_i}^{N} \log \frac{\exp(sim(z_i, z_j)/\tau)}{\sum_{k=1}^{2N} \mathbb{1}_{[(k \neq i) \wedge ((g_k = g_i) \vee (p_k \neq p_i))]} \exp(sim(z_i, z_k)/\tau)} \tag{4}$$

in which $i \in \{1, 2, ..., N\}$ are the indices for the standard batch slices, $j \in \{N+1, N+2, ..., 2N\}$ are the indices for the augmented batch slices, $g_i$ refers to the group associated with slice $i$ in the batch, $p_i$ refers to the patient associated with slice $i$ in the batch, $G$ represents the average number of slices per group (calculated as the total number of unaugmented slices divided by the total number of groups), $z$ is the embedding for a slice in the batch, $\tau$ is a temperature parameter, and $sim$ is a similarity function which was cosine similarity in this study. The formula shares similarities with NT-Xent loss but introduces some modifications:

1. There are two summations which reflects all the group slices for a particular data point.
2. In the numerator of the logarithmic term, all the group slices for particular data point are considered similar, encouraging the development of similar embeddings for these slices.

3. The denominator excludes patient slices for a particular data point that are not part of the same group

Excluding non-group patient slices ensures that the model does not promote dissimilarity between non-group slices from the same patient. This ensures that we can sum multiple group losses together without counteracting their effects. In our study, we considered patient, volume, adjacent slice group contrastive losses in addition to NT-Xent loss. Our overall loss formulation is as follows:

$$\mathcal{L}_{\text{contrastive}} = \mathcal{L}_{\text{NT-Xent}} + \lambda_1 \mathcal{L}_{\text{patient group}} + \lambda_2 \mathcal{L}_{\text{volume group}} + \lambda_3 \mathcal{L}_{\text{slice group}} \tag{5}$$

where $\mathcal{L}_{\text{patient group}}, \mathcal{L}_{\text{volume group}}, \mathcal{L}_{\text{slice group}}$ are the group contrastive losses associated with patient, volume, and adjacent slice groups respectively and $\{\lambda_1, \lambda_2, \lambda_3\}$ are constants. The summation of different contrastive losses in our overall loss $\mathcal{L}_{\text{contrastive}}$ is the focus of the ablation study in Section 4.6.

**Batch sampler** We employ $\mathcal{L}_{\text{contrastive}}$ to train a SimCLR network (35) using a ResNet-18 encoder (24), which is our $g_\phi$. In practice, standard random data loading would yield minimal impact from $\mathcal{L}_{\text{contrastive}}$ due to the low probability of randomly selecting two slices from the same group (e.g., same patient, same volume, or adjacent slices) within a batch. To address this, we introduce a batch sampler designed to increase their occurrence. The batch sampling process can be summarized as follows: 1. create a singular slice group for all the dataset slices, 2. for each group, randomly include a slice from the same patient for each of the groupings used (e.g., patient and volume), 3. combine groups from different patients to form a batch. An epoch consists of all the dataset groups: thus, more contrastive loss groups result in larger epochs. The batch sampler would be reset every epoch to ensure randomness during training. Please see Appendix C for more details.

Training the SimCLR network with this batch sampler eliminates the need for complex algorithmic adjustments to accommodate multiple contrastive loss groups, a challenge in other AL contrastive learning methodologies (85). We conduct the training over 100 epochs using an ADAM optimizer, with a learning rate of 3e-4, a weight decay of 1.0e-6, and a batch size of 8 for one or three groups and 9 for two groups.

### 3.2.2 Segmentation model training

Once we have $g_\phi$, we can calculate $d_\phi$ and collect annotations for unlabeled slices to train our segmentation model. Our AL evaluation consists of several rounds of annotation collection from an oracle. After each round of annotation collection, we train five segmentation models and record the model's test score with the highest validation score. We repeat the AL experiment five times for each algorithm, each time with a different random seed, and report the average model test score per round. For weakly-supervised segmentation, we train a Dual-Branch Network with Dynamically Mixed Pseudo Labels Supervision (DMPLS) (40), which reported strong metrics on the ACDC dataset using weak supervision. For full supervision, we train UNet (58) which is a frequently used fully-supervised segmentation baseline model. We calculate the 3D DICE score on each 3D volume and report the average on all the volumes in the test set. For full supervision, we also provide results using a pre-trained segmentation model with a ResNet-50 backbone (42) (pre-trained on medical images (42) for medical datasets and ImageNetV2 for non-medical datasets). We do not report weakly-supervised results on pre-trained architectures because the weakly-supervised architectures cannot easily utilize pre-trained backbones.

## 4 Experiments

### 4.1 Datasets

**ACDC (Automated Cardiac Diagnosis Challenge)** The ACDC dataset (4) consists of 200 short-axis cine-MRI scans from 100 patients in the training set and 100 scans from 50 patients in the test set. Acquired data was fully anonymized and handled within the regulations set by the local ethical committee of the Hospital of Dijon (France). Each patient has two annotated end-diastolic (ED) and end-systolic (ES) phase scans. Annotations are available for three structures: the right ventricle (RV), myocardium (Myo), and left ventricle (LV). Additionally, scribble annotations have been provided for each scan by a previous study (72). The training set size consists of 1448 slices.

**CHAOS (Combined Healthy Abdominal Organ Segmentation)**   The CHAOS dataset (33) comprises of abdominal CT images from 20 subjects in the training set, primarily used for liver and vessel segmentation. The anonymized dataset was collected from the Department of Radiology, Dokuz Eylul University Hospital, Izmir, Turkey and the study was approved by the Institutional Review Board of Dokuz Eylul University. Each patient's CT scans contains approximately 144 slices. Binary segmentation masks for the liver are provided. We resampled, cropped, and normalized the images, following the process described in a previous study (72). We partitioned the training set into training (75%), validation (10%), and test (15%) subsets. The training set size consists of 2351 slices.

**MS-CMR (Multi-sequence Cardiac MR Segmentation Challenge)**   The MS-CMR dataset (22; 89; 78) contains late gadolinium enhancement (LEG) MRI images from 45 patients who underwent cardiomyopathy. The data has been collected from Shanghai Renji hospital with institutional ethics approval and has been anonymized. These images were multi-slice, acquired in the ventricular short-axis views. We obtained realistic and manual scribble annotations from a prior study (84). Similar to this study (84), we split the data such that 25 patients were assigned to train, 5 to validation, and 20 to test, resulting in 382 slices in the training set. For data processing, we resampled the images into a resolution of 1.37x1.37 mm, and then they were cropped or padded to a fixed size of 212 x 212 pixels. During training, the image pixel values were normalized to zero mean and unit variance.

**DAVIS (Densely Annotated Video Segmentation)**   The DAVIS dataset (50; 51) is a densely annotated video dataset associated with the 2016 DAVIS and 2017 DAVIS Challenge. The dataset collection was partially funded by SNF and human subject data was collected ethically, to the best of our knowledge. In our study, we utilized the train and val sets associated with the 2016 DAVIS Challenge which contained 30 and 20 videos respectively and all frames with 480p resolution. While we used the 2016 DAVIS train set, we created the val and test set by further splitting the 2016 DAVIS val set into 5 and 15 videos respectively. Our train set consisted of 2079 densely annotated frames.

For additional generalizability of our approach, we evaluated our methodology on both weak and full supervision, depending on data availability. Because the CHAOS, MS-CMR, and DAVIS dataset do not contain any hierarchical organization of multiple volumes/videos, we did not use the patient group loss. For video segmentation, in our contrastive loss we treated each video as a "volume" and each video frame as a "slice"; thus, we considered both the volume and adjacent slice group loss.

## 4.2   Experimental settings

When conducting experiments with pre-trained segmentation models, for ACDC, CHAOS, and MS-CMR we collect annotations in cumulative increments of 2%, 3%, 4%, 5%, 10%, 15%, 20%, and 40% in each round. Because segmentation networks require more training data for natural images, for the DAVIS dataset we collect annotations in cumulative increments of 10%, 20% , 30%, and 40% in each round. When conducting experiments with pre-trained segmentation models, because of the benefit of prior pre-training, we collect annotations in cumulative increments of 1%, 2%, 3%, 4%, and 5% in each round.

Because solving the Coreset problem is NP-Hard, we utilized the K-Center Greedy algorithm for our Coreset implementation (48), which is a $2 - OPT$ solution (64) and produces very competitive results in comparison to other more computationally-intensive solutions. We compared our approach to vanilla Coreset (K-Center Greedy) (64), Random Sampling, CoreGCN (7), TypiClust (23; 39), Stochastic Batches (using Deep Ensembles with Entropy) (20), VAAL (66), Deep Ensembles (utilizing Variance Ratio scoring) (3), and Bayesian Deep Learning (utilizing the BALD score) (21). To ensure a fair comparison, all approaches were evaluated using the same experimental settings.

All of our experiments were primarily conducted with a single Tesla V100 GPU on an internal cluster. Our contrastive learning-based encoder and segmentation models consumed approximately 3 GB of GPU memory while training. The contrastive encoder's training speed was approximately 40 slices/second which resulted in a training time of 100 minutes, 200 minutes, and 300 minutes on ACDC for one, two, and three group losses respectively. One single AL experiment for our method on ACDC (eight rounds with five models trained per round) took approximately 24 hours and used about 400 MB of storage.

Table 1: DICE scores for ACDC, MS-CMR, and CHAOS

| | Weakly-supervised | | | | | Fully-supervised | | | | |
|---|---|---|---|---|---|---|---|---|---|---|
| | 2% | 3% | 4% | 5% | 40% | 2% | 3% | 4% | 5% | 40% |
| | | | | | ACDC | | | | | |
| BALD (21) | 44.8 | 54.8 | 61.4 | 66.2 | 86.4 | 53.8 | 66.0 | 67.9 | 71.7 | 89.3 |
| Variance Ratio (3) | 43.3 | 45.0 | 54.2 | 61.4 | 85.9 | 52.6 | 62.2 | 66.6 | 69.1 | 86.9 |
| Random (48) | 44.9 | 45.7 | 59.2 | 66.9 | 86.8 | 66.3 | 76.9 | 79.3 | 80.1 | **90.2** |
| VAAL (66) | 41.8 | 47.8 | 66.1 | 72.1 | 86.4 | 63.2 | 75.2 | 79.5 | 81.3 | 89.9 |
| Coreset (64) | 45.2 | 49.8 | 68.7 | 70.1 | **86.9** | 58.9 | 69.3 | 75.7 | 81.9 | **90.2** |
| TypiClust (23; 39) | 44.8 | 45.6 | 67.6 | 73.1 | 85.7 | 66.6 | 75.8 | 79.5 | 81.7 | 89.5 |
| Stochastic Batches (20) | 36.9 | 39.5 | 53.7 | 57.4 | 85.8 | 60.3 | 69.8 | 74.1 | 75.4 | 89.3 |
| CoreGCN (7) | 40.8 | 49.3 | 69.1 | 74.1 | 86.3 | 67.7 | 74.9 | 79.0 | 80.7 | 89.6 |
| Ours | **52.3** | **59.8** | **73.3** | **76.1** | 86.4 | **70.9** | **77.4** | **81.6** | **82.5** | 90.2 |

| | Weakly-supervised | | | | | Fully-supervised | | | | |
|---|---|---|---|---|---|---|---|---|---|---|
| | 2% | 3% | 4% | 5% | 40% | 2% | 3% | 4% | 5% | 40% |
| | | | | MS-CMR | | | | CHAOS | | |
| BALD | 37.0 | 48.9 | 57.4 | 60.0 | 85.8 | 79.6 | 80.7 | 80.4 | 81.1 | 95.8 |
| Variance Ratio | 41.3 | 46.8 | 52.8 | 55.3 | 84.3 | 74.9 | 72.9 | 76.6 | 76.2 | 92.8 |
| Random | 39.7 | 55.0 | 61.0 | 61.5 | 85.7 | **80.7** | 81.7 | 84.2 | 85.1 | 96.2 |
| Coreset | 28.7 | 53.6 | 56.9 | 58.7 | **86.7** | 80.0 | 80.4 | 81.2 | 88.1 | **96.5** |
| Stochastic Batches | 38.5 | **56.2** | 59.8 | 60.5 | 86.2 | 77.2 | **82.9** | 83.8 | 84.7 | 96.1 |
| CoreGCN | 27.0 | 48.7 | 57.1 | 57.2 | 85.5 | 67.8 | 77.7 | 77.8 | 74.4 | 94.3 |
| Ours | **44.3** | 53.3 | **63.4** | **63.5** | 86.3 | 80.5 | 82.5 | **85.9** | **90.3** | 96.3 |

Table 2: DICE scores for DAVIS

| | Fully-supervised | | | | |
|---|---|---|---|---|---|
| | 10% | 20% | 30% | 40% | Mean |
| BALD | **43.6** | 42.0 | 43.4 | 43.8 | 43.1 |
| Variance Ratio | 36.1 | 31.2 | 34.9 | 40.7 | 35.6 |
| Random | 39.6 | 40.5 | **47.4** | **48.5** | 45.5 |
| Coreset | 31.7 | 39.4 | 42.2 | 42.1 | 41.2 |
| Stochastic Batches | 40.3 | 41.5 | 45.1 | 47.6 | 44.7 |
| Ours | 42.8 | **45.2** | 45.5 | 46.6 | **45.8** |

## 4.3 Results

Tables 1 and 2 summarize the results from our experiments on weak and full annotations for the ACDC, MS-CMR, CHAOS, and DAVIS datasets when trained from scratch. Our method excels in low annotation budget scenarios (2%-5%) on the ACDC dataset, outperforming other methods by up to 10% in some cases. This advantage is vital in the medical field where annotation costs are often high. On the MS-CMR, CHAOS, and DAVIS datasets, our method remains highly competitive,

Table 3: Mean DICE scores over all annotation datapoints with pre-trained weights

| | Fully-supervised | | |
|---|---|---|---|
| | ACDC | CHAOS | DAVIS |
| Random | 78.4 | 94.9 | 74.9 |
| Stochastic Batches | 77.3 | 95.0 | **75.1** |
| Coreset | 78.6 | 95.1 | 74.6 |
| Ours | **79.5** | **95.2** | **75.1** |

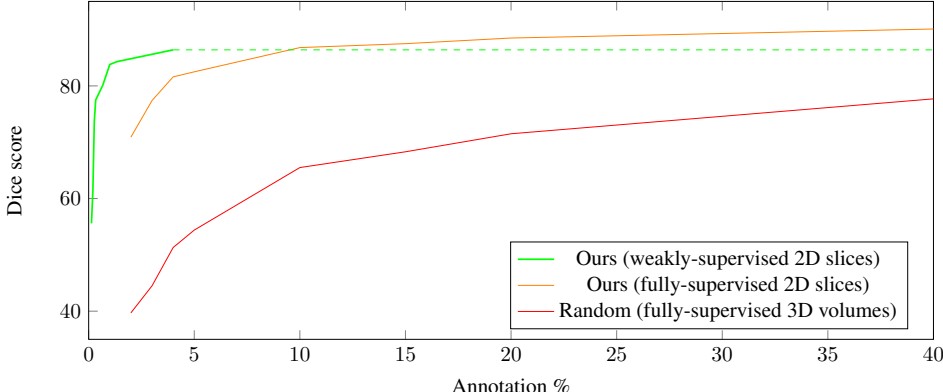

Figure 2: Describes the relationship between model performance and annotation time for our method utilizing weakly and fully-supervised 2D slices and random sampling of fully-supervised 3D volumes on the ACDC dataset. Annotation % is measured as the percentage of the fully-labeled ACDC training data. For weak supervision, we extrapolate the percentage of fully-labeled data based on equivalent annotation time (we follow prior work which assumes that annotators annotate scribbles 15x as fast as the full masks (72)). The dashed green line represents the performance of our method using weakly-supervised 2D slices with 40% of the ACDC training data.

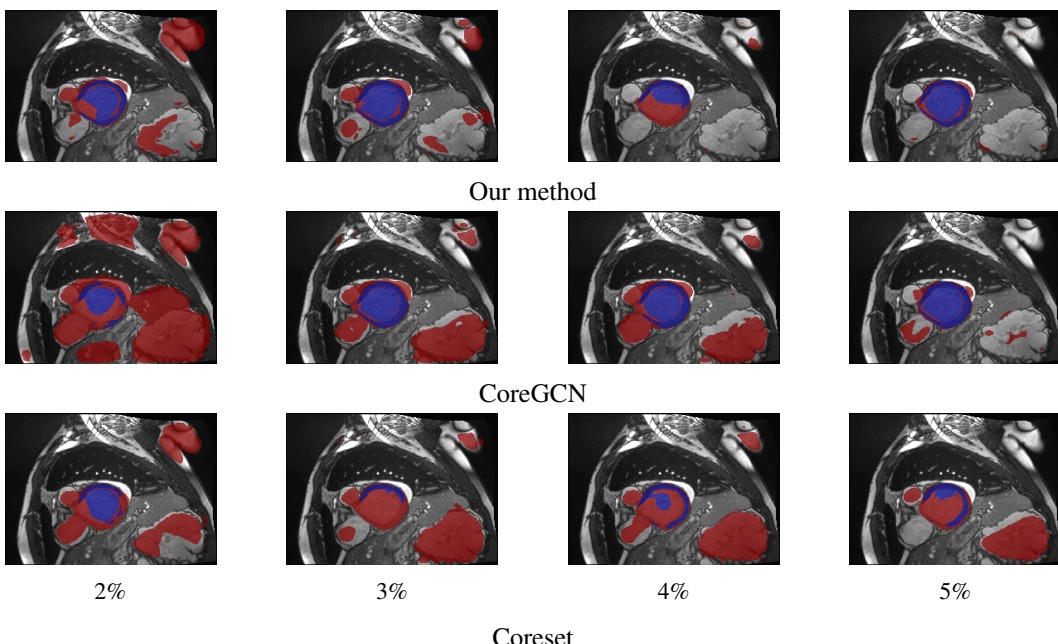

Figure 3: Qualitative comparison of our method, CoreGCN, and Coreset. Blue indicates agreement between model predictions and groundtruth masks and red indicates disagreement.

consistently achieving the highest or close to the highest performance for a particular annotation level.

Additionally, our algorithm consistently demonstrates superior performance in both weak and full annotation scenarios, unlike other top-performing methods which struggle in one of these settings. Compared to other algorithms, our method shows superior performance on different datasets as well.

We note that clinically acceptable DICE scores for similar medical segmentation tasks range from 0.5-0.9, depending the task (68; 25; 75; 8). However, even lower DICE scores can be clinically useful, especially for particular volumes with higher scores or in conjunction with semi-automated segmentation methods (57).

Table 4: Ablation study based on the mean DICE scores for the 2-5% weak annotation datapoint

| Coreset | NT-Xent | Patient | Volume | Slice | mDICE |
|:---:|:---:|:---:|:---:|:---:|:---:|
| ✓ | | | | | 58.5 |
| ✓ | ✓ | | | | 59.5 |
| ✓ | | ✓ | | | 60.0 |
| ✓ | | | ✓ | | **60.7** |
| ✓ | | | | ✓ | 58.1 |
| ✓ | | ✓ | ✓ | | 61.5 |
| ✓ | | | ✓ | ✓ | 60.9 |
| ✓ | ✓ | | ✓ | | **62.6** |
| ✓ | ✓ | ✓ | ✓ | | **65.4** |
| ✓ | | ✓ | ✓ | ✓ | 64.1 |
| ✓ | ✓ | ✓ | ✓ | ✓ | 61.1 |

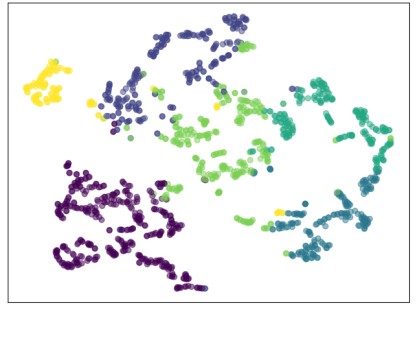

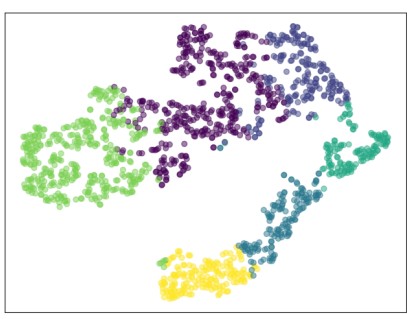

(a) NT-Xent Loss          (b) Our Loss

Figure 4: t-SNE visualization of dataset clusters generated by different $g_\phi$

Our fully-supervised experiments with pre-trained models are in Table 3. We saw improvements with our method in comparison to the best performing comparison methods. Please refer to Appendix D for comprehensive results across all datasets, including calculated bootstrapping standard errors.

## 4.4 Relationship between model performance and annotation time

In Figure 2 we provided a graph which describes the relationship between model performance and annotation time for our method utilizing weakly and fully-supervised 2D slices and random sampling of fully-supervised 3D volumes on the ACDC dataset. To ensure a fair comparison between the different methods, we do not incorporate any of the results from the pre-trained segmentation models. For the 3D results, similar to the 2D U-Net, we train a 3D U-Net from scratch. Given comparable annotation time, our methods trained on both weakly and fully-supervised 2D slices far exceed the performance of random sampling of 3D volumes and achieve 3D volume maximum performance (with the given budget) with much less annotation time.

## 4.5 Comparison with related work

Of the diversity-based comparison methods (Coreset, VAAL, TypiClust, Random), the performance for these three are generally worse than our method. For example, our method achieves the best performance on 21 out of 27 comparison points (Tables 1, 2, 3). The next closest is Random sampling, which achieves the best performance on 5 out of 27 comparison points. Of the entropy-based methods (BALD, Variance Ratio, Stochastic Batches), Stochastic Batches achieves the best performance on 4 out of 27 comparison points, the best out of the group.

Coreset and CoreGCN share similarities with our method. However, on almost all the comparison points, they perform much worse. In Figure 3, there is a qualitative comparison of our method, CoreGCN, and Coreset on a difficult slice in the volume after several weak annotation rounds. With more requested annotations, our method is able to reduce errors even in difficult slices. With 5% weak annotation, while Coreset and CoreGCN still retain large error artifacts, our method has minimally visible errors.

## 4.6 Ablation study

In Table 4, we see our ablation experiments. The first three sections represent our experiments with one, two, and three or more contrastive losses respectively. We note that all the contrastive loss experiments perform better than vanilla Coreset. Additionally, generally the larger combination of losses tend to outperform the smaller combination of losses. The best loss combination involves the volume group loss, patient group loss, and NT-Xent.

One interesting observation is that the loss associated with the volume group — the most diverse group — tends to produce the best additive performance and the loss associated with the the adjacent slice group — the least diverse group — tends to have the worst additive performance. We see this trend in the one loss experiments as well in the two loss and three or more loss experiments, where the combinations with the volume group loss tend to produce the best performance and the combinations with the adjacent slice group loss tends to produce the worst performance.

In order to visualize how effective the learned $g_\phi$ trained with different losses are for Coreset, we applied k-means clustering to the generated dataset features with a contrastive encoder trained with NT-Xent and trained with our optimal loss and visualized the quality of cluster labels using a t-SNE plot, which can be seen in Figure 4. We note that the clusters from our loss show good cohesion (tightly grouped), separation between clusters, and are easy to differentiate. However, the clusters from NT-Xent show much less cohesion (points are spread over more space) and the separation is less defined. Higher quality clustering emphasizes points are well separated which leads to better performance when trying to find representative points using Coreset.

## 5 Conclusion

In our research, we introduced a novel metric learning method for Coreset to perform slice-based active learning in 3D medical segmentation. By leveraging diverse data groups in our feature representation, we were able to learn a metric that promoted diversity and our Coreset implementation was able to outperform all existing methods on medical and non-medical datasets in weak and full annotation scenarios with a low annotation budget. Due to limited computational resources, we restricted the number of experiment runs and models we trained. We also acknowledge that we did not fully consider training set bias which can result in unfair outcomes for underrepresented groups. In future work, we hope to remedy some of these issues and focus more robustly on the applications of our approach in diverse domains.

## Acknowledgments

This work was partially supported by NSF 2211557, NSF 1937599, NSF 2119643, NSF 2303037, NSF 2312501, NASA, SRC JUMP 2.0 Center, Amazon Research Awards, and Snapchat Gifts.

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

## A  Loss weights

In our experiments and ablation study, a 1.0 weight was applied whenever the NT-Xent loss was used. Additionally, a 1.0 weight was applied if only one group contrastive loss was used. In the ablation study on the ACDC dataset, for the experiments with multiple group contrastive losses, we tried different combinations of weights for the group contrastive losses and reported the best results. We will utilize the formulation from Equation 5 for the loss weights and report them as a four-tuple $(a, b, c, d)$ which corresponds to $(\lambda_0, \lambda_1, \lambda_2, \lambda_3)$ (in which $\lambda_0$ is 1 if the NT-Xent loss was used and 0 if it is not). The combinations we tried are as follow (with the best bolded):

- patient and volume group loss without NT-Xent loss: (0, 0.125, 0.875, 0), **(0, 0.50, 0.50, 0)**
- volume and slice group loss without NT-Xent loss: (0, 0, 0.125, 0.875), **(0, 0, 0.50, 0.50)**
- patient and volume group loss with NT-Xent loss: (1,0. 10 , 0.35, 0), (1, 0.117, 0.233, 0), **(1, 0.05, 0.35, 0)**
- patient, volume, and slice group loss without NT-Xent loss: (0, 0.05, 0.25, 0.7), **(0, 0.33, 0.33, 0.33)**
- patient, volume, and slice group loss with NT-Xent loss: (1, 0.10, 0.20, 0.05), **(1, 0.05, 0.35, 0.025)**

We utilized the best loss/weight combination for our ACDC experiments. For the CHAOS, MSCMR, and DAVIS datasets, because there is only one volume per patient and thus no difference between the patient and volume loss, we tested two loss/weight combinations:

- volume loss with weight 0.35 with NT-Xent loss
- volume loss with weight 0.10 and slice loss with weight 0.30 with NT-Xent loss

Both weight combinations performed better than other comparison methods though volume loss with weight 0.35 with NT-Xent loss performed slightly better than the other combination.

For the pre-trained weights, we found that best results were obtained on the ACDC dataset with just the patient group without the NT-Xent loss. We tested different combinations of groups but found the results were worse. We used the same weight setting for CHAOS and DAVIS

## B  Theoretical bounds for loss

First, we assume that the expectation over the data distribution of the volume-based loss and slice-based loss are equivalent. Formally, this is described as follows:

$$E_{\mathbf{x},\mathbf{y}\sim p_{\mathcal{Z}}}[\mathcal{L}_{volume}(\hat{\mathbf{y}}, \mathbf{y})] = E_{\mathbf{x}',\mathbf{y}'\sim p_{\mathcal{Z}'}}[\mathcal{L}_{slice}(\hat{\mathbf{y}}', \mathbf{y}')]$$

where $\mathcal{L}_{volume} : \mathcal{X} \times \mathcal{Y} \to \mathbb{R}$ and $\mathcal{L}_{slice} : \mathcal{X}' \times \mathcal{Y}' \to \mathbb{R}$ are the volume-based and slice-based loss respectively. For the rest of the proof, $\mathcal{L}_{slice}(\cdot)$ is referred to by $\mathcal{L}(\cdot)$. Following the derivation provided in the original Coreset paper (64) we have:

$$
\begin{aligned}
E_{\mathbf{x}',\mathbf{y}'\sim p_{\mathcal{Z}'}}[\mathcal{L}(\hat{\mathbf{y}}', \mathbf{y}')] \leq & \underbrace{\left| E_{\mathbf{x}',\mathbf{y}'\sim p_{\mathcal{Z}'}}[\mathcal{L}(\hat{\mathbf{y}}', \mathbf{y}')] - \frac{1}{n}\sum_{i\in[n]}\mathcal{L}(\hat{\mathbf{y}}', \mathbf{y}') \right|}_{\text{Generalization Error}} + \underbrace{\frac{1}{|\mathbf{s}|}\sum_{j\in\mathbf{s}}\mathcal{L}(\hat{\mathbf{y}}', \mathbf{y}')}_{\text{Training Error}} \\
& + \underbrace{\left| \frac{1}{n}\sum_{i\in[n]}\mathcal{L}(\hat{\mathbf{y}}', \mathbf{y}') - \frac{1}{|\mathbf{s}|}\sum_{j\in\mathbf{s}}\mathcal{L}(\hat{\mathbf{y}}', \mathbf{y}'), \right|}_{\text{Core-Set Loss}} \\
\leq & \underbrace{\left| \frac{1}{n}\sum_{i\in[n]}\mathcal{L}(\hat{\mathbf{y}}', \mathbf{y}') - \frac{1}{|\mathbf{s}|}\sum_{j\in\mathbf{s}}\mathcal{L}(\hat{\mathbf{y}}', \mathbf{y}'), \right|}_{\text{Core-Set Loss}}
\end{aligned}
$$

We get the last line of the inequality because we assume the Training Error is zero and, similar to the original Coreset paper, we assume the Generalization Error is zero, (which is a reasonable assumption because most CNNs have very small generalization error) (64).

Thus, our active learning objective can be re-defined as:

$$\underset{s^1 \subset D, |s^1| \leq k}{\operatorname{argmin}} \frac{1}{n} \sum_{i \in [n]} \mathcal{L}(\hat{\mathbf{y}}', \mathbf{y}') - \frac{1}{|\mathbf{s}|} \sum_{j \in \mathbf{s}} \mathcal{L}(\hat{\mathbf{y}}', \mathbf{y}') \tag{6}$$

We will now present the following theorem:

**Theorem 1.** *Given $n$ i.i.d. samples drawn from $p_{\mathcal{Z}'}$ as $\{\mathbf{x}'_i, \mathbf{y}'_i\}_{i \in [n]}$, and set of points $\mathbf{s}$. If loss function $\mathcal{L}(\hat{\mathbf{y}}', \mathbf{y}')$ is $\lambda^l$-Lipschitz continuous for all $\hat{\mathbf{y}}', \mathbf{y}'$ and bounded by $L$, segmentation function $\eta_{\mathbf{c}}(\mathbf{x}') = p(\mathbf{y}' = \mathbf{c}|\mathbf{x}')$ is $\lambda^\eta$-Lipshitz continuous for all $\mathbf{x}' \in \Omega'$ and $\mathbf{c} \in \mathcal{Y}'$, $\mathbf{s}$ is $\delta$ cover of $\{\mathbf{x}'_i, \mathbf{y}'_i\}_{i \in [n]}$, and $l(\hat{\mathbf{y}}'_{s(j)}, \mathbf{y}'_{s(j)}) = 0 \quad \forall j \in [m]$; with probability at least $1 - \gamma$,*

$$\frac{1}{n} \sum_{i \in [n]} \mathcal{L}(\hat{\mathbf{y}}'_i, \mathbf{y}'_i) - \frac{1}{|\mathbf{s}|} \sum_{j \in \mathbf{s}} \mathcal{L}(\hat{\mathbf{y}}'_j, \mathbf{y}'_j) \leq \delta(\lambda^l + \lambda^\eta L 2^{hwdk}) + \sqrt{\frac{L^2 \log(1/\gamma)}{2n}}.$$

Similar to (64), we can start our proof with bounding $E_{\mathbf{y}'_i \sim \eta(\mathbf{x}'_i)} \mathcal{L}(\hat{\mathbf{y}}'_i, \mathbf{y}'_i)$. Following a similar approach to (64), we get:

$$E_{\mathbf{y}'_i \sim \eta(\mathbf{x}'_i)} \mathcal{L}(\hat{\mathbf{y}}'_i, \mathbf{y}'_i) \leq \delta(\lambda^l + \lambda^\eta L 2^{hwdk})$$

Again, following (64), we can use the Hoeffding's Bound to conclude the rest of the proof.

## C  Batch sampler

The pseudocode for the batch sampler can be seen in Algorithm 1. This implementation assumes the use of all three group contrastive losses. If the adjacent slice group is removed, then remove line 6. If the volume group is removed, then remove line 7. If the patient group is removed, then remove line 8. An illustration on how the batch sampler works can be seen in Figure 5

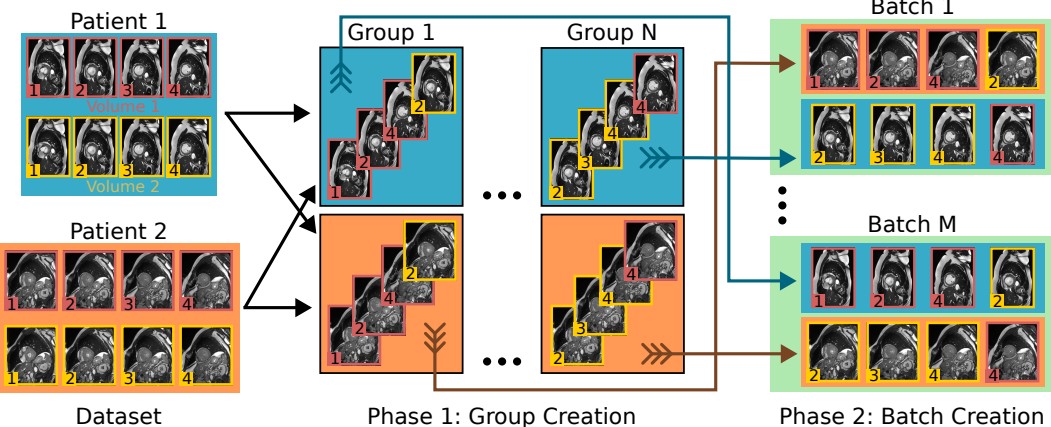

Figure 5: Overview of the batch sampler for Group-based Contrastive Learning

**Algorithm 1** Batch sampler for one epoch

---

**Input:** List of patient groups $P_1, P_2, ..., P_L$. Each patient group $i$ contains a list of volume groups $V_1^i, V_2^i, ..., V_K^i$. Each volume group $j$ for patient $i$ contains a list of slices $s_1^{ij}, s_2^{ij}, ..., s_T^{ij}$. Batch size is $M$.

1:  $PG \leftarrow list()$
2: **for all** $P_1, P_2, ..., P_L$ **do**
3:     $groups \leftarrow ()$
4:     **for all** $V_1^i, V_2^i, ..., V_K^i$ **do**
5:         **for all** $s_1^{ij}, s_2^{ij}, ..., s_T^{ij}$ **do**
6:             $s \leftarrow RandomChoice(s_{k-1}^{ij}, s_{k+1}^{ij})$
7:             $v \leftarrow RandomChoice(\cup_{t \neq k} s_t^{ij})$
8:             $p \leftarrow RandomChoice(\cup_{V \in P_i : s \in V} s / \{s_j^{ij}\})$
9:             $group \leftarrow [s_k^{ij}, s, v, p]$
10:           Add $group$ to $groups$
11:         **end for**
12:     **end for**
13:     Add $groups$ to $PG$
14: **end for**
15: $batches \leftarrow ()$
16: **while** $|PG| \geq M$ **do**
17:     $batch \leftarrow list()$
18:     $S \leftarrow ()$
19:     **while** $|batch| < M$ **do**
20:         $groups \leftarrow RandomChoice(\cup_{g \in PG \wedge g \notin S})$
21:         $group \leftarrow RandomChoice(groups)$
22:         $groups \leftarrow groups \backslash group$
23:         **if** $|groups| = 0$ **then**
24:             $PG \leftarrow PG \backslash groups$
25:         **else**
26:             Add $groups$ to $S$
27:         **end if**
28:         Extend $batch$ with $group$
29:     **end while**
30:     Add $batch$ to $batches$
31: **end while**
32: **return** batches

---

# D   Additional results

We have included all collected results on all the datasets. We also report twice the bootstrap standard error for the means over the different experiment runs. Our process is as follows. We generate 1000 bootstraps. For each bootstrap, we generate a resampling with replacement of the test set volumes and generate the mean of the test set metrics (average 3D Dice score over the test set) over the different experiment runs at the desired evaluation point. We then calculate the sample standard deviation of the bootstrapped means and multiply by two.

Because the CHAOS and DAVIS have fewer volumes (and the bootstrapping errors are artifically inflated), rather generating a resampling with replacement on the test set volumes, we generate a resampling over the slices.

Table 5: DICE scores for ACDC

| | Weakly-supervised | | | |
|---|---|---|---|---|
| | 2% | 3% | 4% | 5% |
| BALD | 44.8±1.9 | 54.8±1.9 | 61.4±1.8 | 66.2±1.6 |
| Variance Ratio | 43.3±2.1 | 45.0±2.2 | 54.2±2.4 | 61.4±2.1 |
| Random | 44.9±2.3 | 45.7±2.2 | 59.2±2.1 | 66.9±1.9 |
| VAAL | 41.8±2.3 | 47.8±2.2 | 66.1±2.1 | 72.1±1.9 |
| Coreset | 45.2±2.1 | 49.8±2.0 | 68.7±2.0 | 70.1±1.8 |
| TypiClust | 44.8±2.2 | 45.6±2.0 | 67.6±1.8 | 73.1±1.6 |
| Stochastic Batches | 36.9±2.2 | 39.5±2.0 | 53.7±1.8 | 57.4±1.5 |
| CoreGCN | 40.8±2.2 | 49.3±2.0 | 69.1±1.9 | 74.1±1.7 |
| Ours | 55.6±2.1 | 61.4±1.9 | 73.7±2.2 | 77.5±1.6 |
| | 10% | 15% | 20% | 40% |
| BALD | 77.7±1.3 | 81.6±1.2 | 83.3±1.1 | 86.4±0.9 |
| Variance Ratio | 74.3±1.9 | 83.2±1.5 | 84.4±1.2 | 85.9±0.9 |
| Random | 76.5±1.4 | 83.0±1.3 | 84.1±1.1 | 86.7±0.9 |
| VAAL | 79.6±1.4 | 82.4±1.2 | 84.4±1.1 | 86.4±0.9 |
| Coreset | 80.3±1.3 | 83.6±1.2 | 85.3±1.1 | 86.9±1.0 |
| TypiClust | 77.7±1.4 | 82.0±1.3 | 83.6±1.3 | 85.7±1.2 |
| Stochastic Batches | 69.2±1.4 | 78.5±1.3 | 81.9±1.2 | 85.8±1.1 |
| CoreGCN | 78.4±1.3 | 82.9±1.3 | 83.9±1.1 | 86.3±0.9 |
| Ours | 80.1±1.4 | 83.8±1.3 | 84.3±1.2 | 86.4±1.0 |
| | Fully-supervised | | | |
| | 2% | 3% | 4% | 5% |
| BALD | 53.8±3.7 | 66.0±2.7 | 67.9±2.5 | 71.7±2.2 |
| Variance Ratio | 52.6±3.3 | 62.2±3.2 | 66.6±3.6 | 69.1±3.3 |
| Random | 66.3±3.2 | 76.9±3.0 | 79.3±2.4 | 80.1±1.8 |
| VAAL | 63.2±2.6 | 75.2±2.3 | 79.5±2.1 | 81.3±1.8 |
| Coreset | 58.9±3.7 | 69.3±3.3 | 75.7±3.6 | 81.9±2.7 |
| TypiClust | 66.6±3.4 | 75.8±3.1 | 79.5±2.4 | 81.7±2.1 |
| Stochastic Batches | 60.3±3.0 | 69.8±2.8 | 74.1±2.6 | 75.4±2.5 |
| CoreGCN | 67.7±2.6 | 74.9±2.4 | 79.0±2.2 | 80.7±2.1 |
| Ours | 70.9±2.7 | 77.4±2.4 | 81.6±2.2 | 82.5±2.1 |
| | 10% | 15% | 20% | 40% |
| BALD | 81.6±1.8 | 85.5±1.2 | 85.9±1.0 | 89.3±0.9 |
| Variance Ratio | 76.4±3.2 | 80.7±3.0 | 83.6±2.4 | 86.9±1.3 |
| Random | 86.8±1.6 | 88.1±1.3 | 88.3±1.1 | 90.2±0.9 |
| VAAL | 85.9±1.4 | 88.0±1.3 | 87.9±1.2 | 89.9±1.1 |
| Coreset | 85.8±2.2 | 87.7±1.3 | 88.8±1.1 | 90.2±1.0 |
| TypiClust | 85.9±1.3 | 87.1±1.2 | 88.0±1.0 | 89.5±0.9 |
| Stochastic Batches | 81.6±2.4 | 84.6±1.4 | 86.5±1.3 | 89.3±1.1 |
| CoreGCN | 85.9±1.3 | 87.9±1.2 | 88.5±1.2 | 89.6±1.0 |
| Ours | 86.8±1.8 | 87.5±1.4 | 88.5±1.3 | 90.2±1.0 |

Table 6: DICE scores for MSCMR

| | Weakly-supervised | | | |
|---|---|---|---|---|
| | 2% | 3% | 4% | 5% |
| BALD | 37.0±2.8 | 48.9±2.9 | 57.4±2.7 | 60.0±3.0 |
| Variance Ratio | 41.3±5.0 | 46.8±5.1 | 52.8±5.5 | 55.3±3.8 |
| Random | 39.7±3.7 | 55.0±4.0 | 61.0±4.1 | 61.5±3.1 |
| Coreset | 28.7±5.0 | 53.6±4.5 | 56.9±4.1 | 58.7±3.6 |
| Stochastic Batches | 38.5±5.6 | 56.2±4.2 | 59.8±4.6 | 60.5±3.6 |
| CoreGCN | 27.0±5.6 | 48.7±5.3 | 57.1±5.2 | 57.2±3.6 |
| Ours | 44.3±5.2 | 53.3±4.9 | 63.4±5.4 | 63.5±4.5 |
| | 10% | 15% | 20% | 40% |
| BALD | 77.0±2.8 | 79.5±3.2 | 80.1±3.0 | 85.8±2.3 |
| Variance Ratio | 65.8±3.3 | 72.9±3.1 | 78.4±2.9 | 84.3±2.6 |
| Random | 76.0±2.9 | 82.4±3.1 | 83.2±3.0 | 85.7±2.8 |
| Coreset | 74.4±2.8 | 81.4±2.1 | 83.1±2.5 | 86.7±1.9 |
| Stochastic Batches | 76.1±3.2 | 80.6±2.8 | 83.1±2.2 | 86.2±1.9 |
| CoreGCN | 71.0±3.6 | 80.9±3.7 | 83.0±3.8 | 85.5±3.8 |
| Ours | 77.2±2.9 | 82.4±2.3 | 83.6±2.1 | 86.3±2.3 |

Table 7: DICE scores for CHAOS

| | Fully-supervised | | | |
|---|---|---|---|---|
| | 2% | 3% | 4% | 5% |
| BALD | 79.6±1.8 | 80.7±1.8 | 80.4±1.8 | 81.1±1.9 |
| Variance Ratio | 74.9±2.5 | 72.9±2.7 | 76.6±2.6 | 76.2±2.6 |
| Random | 80.7±1.9 | 81.7±1.7 | 84.2±1.5 | 85.1±1.3 |
| Coreset | 80.0±1.8 | 80.4±2.0 | 81.2±1.9 | 88.1±1.0 |
| Stochastic Batches | 77.2±2.3 | 82.9±1.9 | 83.8±1.8 | 84.7±2.0 |
| CoreGCN | 67.8±2.1 | 77.7±2.1 | 77.8±1.2 | 74.4±1.5 |
| Ours | 80.5±1.8 | 82.5±1.5 | 85.9±0.8 | 90.3±1.3 |
| | 10% | 15% | 20% | 40% |
| BALD | 92.6±0.6 | 94.5±0.5 | 94.9±0.5 | 95.8±0.4 |
| Variance Ratio | 82.6±2.4 | 85.1±2.0 | 87.2±1.9 | 92.8±1.2 |
| Random | 92.7±0.8 | 94.3±0.6 | 94.8±0.6 | 96.2±0.3 |
| Coreset | 92.2±0.9 | 94.9±0.5 | 95.8±0.4 | 96.5±0.2 |
| Stochastic Batches | 92.1±1.1 | 93.0±1.0 | 94.1±0.7 | 96.1±0.3 |
| CoreGCN | 85.9±1.0 | 93.3±0.6 | 94.1±0.5 | 94.3±0.3 |
| Ours | 92.5±0.7 | 94.4±0.7 | 95.6±0.5 | 96.3±0.3 |

Table 8: DICE scores for DAVIS

| | Fully-supervised | | | |
|---|---|---|---|---|
| | 10% | 20% | 30% | 40% |
| BALD | 43.6±0.6 | 42.0±0.5 | 43.4±0.4 | 43.8±0.6 |
| Variance Ratio | 36.1±0.5 | 31.2±0.4 | 34.9±0.6 | 40.7±0.3 |
| Random | 39.6±0.6 | 40.5±0.7 | 47.4±0.7 | 48.5±0.5 |
| Coreset | 31.7±0.7 | 39.4±0.6 | 42.2±0.6 | 42.1±0.5 |
| Stochastic Batches | 40.3±0.6 | 41.5±0.8 | 45.1±0.6 | 47.6±0.6 |
| Ours | 42.8±0.7 | 45.2±0.6 | 45.5±0.7 | 46.6±0.7 |

Table 9: DICE scores for ACDC (pretrained)

|  | Fully-supervised | | | | |
| --- | --- | --- | --- | --- | --- |
|  | 1% | 2% | 3% | 4% | 5% |
| Random | 61.9±1.8 | 77.8±1.4 | 81.9±1.2 | 85.0±0.9 | 85.6±0.9 |
| Stochastic Batches | 55.2±1.7 | 79.3±1.3 | 82.5±1.2 | 83.4±1.0 | 86.4±1.0 |
| Coreset | 64.0±1.5 | 78.3±1.4 | 81.7±1.3 | 83.7±1.1 | 84.0±1.2 |
| Ours | 66.1±1.9 | 79.4±1.4 | 82.0±1.3 | 84.3±1.0 | 85.9±1.0 |

Table 10: DICE scores for CHAOS (pretrained)

|  | Fully-supervised | | | | |
| --- | --- | --- | --- | --- | --- |
|  | 1% | 2% | 3% | 4% | 5% |
| Random | 91.8±0.5 | 94.6±0.4 | 95.9±0.3 | 96.2±0.2 | 96.4±0.2 |
| Stochastic Batches | 92.4±0.5 | 94.5±0.3 | 95.5±0.3 | 95.9±0.3 | 96.3±0.3 |
| Coreset | 92.3±0.8 | 94.8±0.4 | 95.7±0.2 | 96.3±0.2 | 96.4±0.2 |
| Ours | 93.4±0.3 | 95.1±0.3 | 95.4±0.2 | 96.1±0.2 | 96.1±0.2 |

Table 11: DICE scores for DAVIS (pretrained)

|  | Fully-supervised | | | | |
| --- | --- | --- | --- | --- | --- |
|  | 1% | 2% | 3% | 4% | 5% |
| Random | 71.0±0.5 | 71.2±0.5 | 76.7±0.6 | 76.2±0.7 | 79.3±0.7 |
| Stochastic Batches | 69.1±0.6 | 73.5±0.7 | 76.9±0.7 | 76.8±0.7 | 78.9±0.6 |
| Coreset | 68.5±0.6 | 73.2±0.6 | 76.9±0.6 | 76.9±0.7 | 77.4±0.6 |
| Ours | 71.1±0.6 | 73.5±0.7 | 76.4±0.6 | 77.2±0.7 | 77.2±0.5 |

