# OpenReview forum: "Integrating Deep Metric Learning with Coreset for Active Learning in 3D Segmentation"
_NeurIPS.cc/2024/Conference — NeurIPS 2024 poster_

### Official Review · Reviewer_4vB1 · 2024-07-11

**Soundness:** 3
**Presentation:** 2
**Contribution:** 3
**Rating:** 7
**Confidence:** 3

**Summary:**

This paper is concerned with the task of 3D segmentation in medical images. Motivated by the high cost of annotations by radiologists, the authors propose a novel active learning method based on the coresets. Here, instead of using the euclidean metric for defining the similarity between different samples, the authors propose to learn a metric via a contrastive loss. The authors propose a group contrastive loss based on the NT-Xent loss which leverages intrinsic similarities in the data (different patients, different volumes of the same patient, adjacent slices) and evaluate their method both on weakly-supervised (with scribble annotations) and fully-supervised tasks and using three datasets from the medial domain and one video segmentation dataset.

**Strengths:**

The paper is well written and methods to improve medical image segmentation with limited annotations are of high interest to the community. The related work is comprehensive and the authors motivate their work well. Overall methods are sound and the evaluation is enough to support the claims.

**Weaknesses:**

**3D vs. 2D segmentation models**
- To the best of my knowledge 3D models (i.e., *not* slice-based models) generally outperform 2D models in most medical image segmentation tasks [a,b]. While I understand the authors' motivation for using 2D (i.e., slice-based) methods, I find their statements (e.g., L. 36-38, 82-83)
    > There were also inefficiencies in previous studies, such as choosing whole volumes instead of individual slices in 3D segmentation, which increased costs [37]. (L. 82-83)

    surrounding this choice somewhat misleading. I would appreciate a more honest discussion where the authors phrase their focus on slice-based methods less as an advantage and more as a limitation of the present work. They could also briefly discuss potential extensions of their work to 3D segmentation. I suppose that generally, this is possible when the adjacent slice group contrastive loss is removed (potentially one could still consider adjacent 3D patches or something similar). The ablation studies also suggest that the adjacent slice group loss doesn't really help much and the authors did not include it in their final loss formulation (L. 582).

- The restriction to slice-based methods and in particular the slice group contrastive loss imply that the proposed methods would assume the annotations acquired during the active learning to also be slice-based. However, for medical image segmentation it might be much easier to acquire annotations for $N$ slices within the same volume than it is to acquire annotations for $N$ slices across different volumes because readers would usually use some interactive segmentation tool and segmentations are similar for adjacent slices.

**Ablation studies**

In Tab. 3 the authors present an ablation study where they study the effect of different group losses on the overall performance. I find this study very interesting, however:
- It seems that ablation studies where only conducted on a single dataset. If so, on which dataset?
- How comparative are the results, given that including more contrastive losses lead to larger epochs and more update steps by design of the batch sampler? Can the authors clarify?
- How did the authors choose the weighting of each loss term here?

**Typos and minor comments:**

- "NT_Xent loss" -> "NT-Xent loss"
- L. 246: "Training speed for for the [...]"
- Fig. 2 could be improved by just adding a column ground truth segmentation and marking the segmentation by the model in a single color. The blue and red colors can be somewhat confusing. Also, which dataset is this from? I suspect the ACDC dataset? Then, shouldn't there be three classes (RV, LV, Myocardium)?

---
[a] Avesta A, Hossain S, Lin M, Aboian M, Krumholz HM, Aneja S. Comparing 3D, 2.5D, and 2D Approaches to Brain Image Auto-Segmentation. Bioengineering. 2023; 10(2):181. https://doi.org/10.3390/bioengineering10020181

[b] Wasserthal, J., Breit, H. C., Meyer, M. T., Pradella, M., Hinck, D., Sauter, A. W., Heye, T., Boll, D. T., Cyriac, J., Yang, S., Bach, M., & Segeroth, M. (2023). TotalSegmentator: Robust Segmentation of 104 Anatomic Structures in CT Images. Radiology. Artificial intelligence, 5(5), e230024. https://doi.org/10.1148/ryai.230024

**Questions:**

- In  L. 582 the authors write that they did not include the slice group loss in the optimal formulation of their loss but in L. 217-220 it seems they included this term in their experiments for all datasets (also if the slice group loss was not included in training of CHAOS, MS-CMR, DAVIS) this would mean that the authors only used the volume group contrastive loss for all these experiments? Can the authors please clarify this in the manuscript?
- How were the individual group losses weighted?
- In L. 224 the authors mention that segmentation requires more training data for natural images. Could you please provide more details on this or a reference?
- Can you better explain the different groups you have, in particular the patient groups? I suspect that these are different pathology subgroups + healthy group (e.g. in the ACDC dataset we would have 5 patient groups)
- On which dataset is the ablation study in Tab. 3 performed?
- Were reference methods trained for the same number of update steps and on the same amount of training data as the proposed method?
- For the random baseline, did the authors choose samples randomly or weigh samples on a per-volume/per-patient level? Otherwise patients for which more volumes/slices are acquired would overrepresent in the coreset, right?

**Limitations:**

The authors discuss limitations of their method in Sec. 5. Maybe the authors can also discuss the possibility of including other groups (e.g. pathologies, patient age/sex/..., ...), in the future? I would also appreciate a more honest discussion of the limitations and implications of slice-based segmentation (see **3D vs. 2D segmentation models** above)

---

> ### Author Rebuttal · Authors · 2024-08-07
>
> We thank the reviewer for their great feedback. We hope to address the main concerns for the reviewer here.
>
> ***3D vs. 2D segmentation models***
>
> Thank you for your feedback related to this. It’s true that generally 3D models outperform 2D models. However, based on our empirical results, 3D models seem to require more training data. In Figure 5 in the attached PDF, we provide performance and annotation time trade-offs for weakly and fully-supervised slice-based active learning and fully-supervised volume-based active learning. However, in order to provide a fair comparison with the 2D U-Net network, we utilize a simple 3D U-Net work without any advanced features. However, a more advanced 3D segmentation model may provide better performance. In future work, we can look into this more thoroughly.
>
> As you mentioned, we can also adapt our methodology to intelligently annotate 3D volumes vs. 2D slices. Thus, we would likely not utilize the volume and slice-adjacency grouping and, instead, consider patient grouping and other possible groupings like pathology, patient height, patient weight, or modality.
>
> You also mentioned another important point that annotation cost is not uniform among all slices and slices for the same volume tend to be lower cost. We unfortunately did not have the opportunity to consider this in our present work. However, in future work we can consider varied costs in our active learning objective.
>
> ***Ablation study***
>
> Thank you for your feedback related to this. We performed the ablation study on the ACDC dataset. The training time for the contrastive learning encoder scaled based on the number of groupings. Thus, the training time for each subgroup in the ablation study was roughly similar. We have also included the loss weights for the ablation study in the general response. Please let us know if there are any additional questions.
>
> ***Typos and comments***
>
> Thank you for your feedback related to this. We will fix the typos in our camera-ready version. Figure 2 is from the ACDC dataset. Initially, we did not want to have too many colors in the diagram but will make sure all three classes are indicated in our revision.
>
> ***Additional questions***
>
> 1. We have discussed this in the general response in the “Loss weights” section. Please let us know if there are any additional questions.
> 2. We have discussed this in the general response in the “Loss weights” section. Please let us know if there are any additional questions.
> 3. Generally, natural images have much more variability than medical imaging which is why more training data tends to be needed [1]. That being said, utilizing pre-trained models can significantly reduce the need for training data, as can be seen with our results in Table 8 in the attached PDF.
> 4. In our paper, we primarily focused on three groups, starting from most general to most specific: patient, volume, and adjacent slice group. The patient group is all the slices associated with volumes pertaining to a particular patient. The volume group is the slices associated with exactly one volume. The adjacent slice group are all the pairs of slices which are adjacent to one another in a particular volume.
> 5. The ablation study is performed on the ACDC dataset.
> 6. Yes, all training methods were trained for the same number of update steps and on the same amount of training data as the proposed method.
> 7. So, yes, in random sampling we chose samples randomly rather than weigh samples on a per-volume/per-patient level. However, patients had roughly the same number of volumes/slices so this was not a significant issue for us.
>
> ***Discussion on limitations***
>
> 1. For our proposed method and the paper implementation, the definition of “group” refers to a set of 2D slices associated with one patient. However, the definition of group can be extended to include 2D slices associated with multiple patients if a more general hierarchical group is used. If this group is valuable, we could incorporate it into our group contrastive loss. In the denominator of the loss, rather than ignoring non-group slices from the same patient we would ignore non-group slices from the same more general group. We also would need to slightly adapt our batch sampler to address utilizing this group. We can provide more details in the camera-ready version of our paper.
> 2. We discussed 3D vs. 2D segmentation models in the previous section. Please let us know if there are any additional issues or questions.
>
> Thank you again for reading our paper and your help and feedback!
>
> [1] Xu, Y., Shen, Y., Fernandez-Granda, C., Heacock, L., & Geras, K. J. (2024). Understanding differences in applying DETR to natural and medical images. arXiv preprint arXiv:2405.17677.

---

> > ### Comment · Reviewer_4vB1 · 2024-08-09
> > **Thanks for the rebuttal!**
> >
> > I thank the authors for providing a comprehensive rebuttal that addressed most of my concerns and questions. Regarding my previous question about the ablation studies:
> > > Thank you for your feedback related to this. We performed the ablation study on the ACDC dataset. The training time for the contrastive learning encoder scaled based on the number of groupings. Thus, the training time for each subgroup in the ablation study was roughly similar.
> >
> > But if I understand correctly, this means that the results are not comparable between methods trained with different number of groupings. E.g., row 6 is not really comparable to row 11.

---

> > > ### Author Response · Authors · 2024-08-09
> > > **Follow up**
> > >
> > > Thank you for question! It's true that the size of the batches (and consequently the training time) are different dependent on the number of groups. However, we hyper-parameter tested the number of epochs for contrastive encoders trained on different number of groups and we obtained the best performance with 100 epochs regardless of the number of groups. Thus, we feel the results in the ablation study are in fact comparable.

---

> > > > ### Comment · Reviewer_4vB1 · 2024-08-09
> > > > **Thanks, I agree with the authors**
> > > >
> > > > Thanks for the prompt response! Yes, hyperparameter optimization of the number of update steps for each method individually is the best thing to do here and I fully agree with the authors' assessment that the results are then comparable.
> > > >
> > > > In case I didn't just miss it, I would recommend the authors to add this imho important bit of information to the manuscript.

---

> > > > > ### Author Response · Authors · 2024-08-09
> > > > > **Will add to manuscript**
> > > > >
> > > > > Will do! Thanks for suggestion!

---

> > > > > > ### Comment · Reviewer_4vB1 · 2024-08-12
> > > > > >
> > > > > > The authors have addressed my concerns in their rebuttal and during the discussion phase and I will therefore increase my score.

---

> > > > > > > ### Author Response · Authors · 2024-08-12
> > > > > > >
> > > > > > > Thank you!

---

### Official Review · Reviewer_aT8e · 2024-07-11

**Soundness:** 3
**Presentation:** 2
**Contribution:** 2
**Rating:** 5
**Confidence:** 4

**Summary:**

The paper introduces a novel active learning approach based on group-based contrastive learning, aiming to optimize data selection and model efficiency in medical image segmentation tasks. By combining NT_Xent loss with various group contrastive losses, it addresses the Coreset problem in active learning, particularly in handling the diversity and complexity of medical image data. The paper provides detailed descriptions of the method's construction and training process, validated through experiments on multiple classical and novel datasets to demonstrate the model's superiority and robustness under different annotation budgets and data complexities. The research findings hold significant academic and practical significance for enhancing workflows in medical image analysis and have profound implications for advancing machine learning applications in medicine.

**Strengths:**

Firstly, by introducing a group-based contrastive learning approach, the paper significantly enhances model performance in medical segmentation tasks. Secondly, it innovatively combines NT_Xent loss with various group contrastive losses, effectively addressing the Coreset problem in active learning and tackling challenges posed by data diversity and complexity in medical image segmentation. The methodology and experimental design are of high quality, providing clear descriptions of contrastive learning loss applications and their integration within the active learning framework. Experimental results demonstrate the model's strong performance under varying annotation budgets and data complexities, highlighting its relevance for medical image analysis and its potential value to the NeurIPS community.

**Weaknesses:**

1. The description is confusing. For instance, the first chapter mentions two main contributions, but the abstract and methods sections also discuss the loss function, which is not explained in the contributions of the first chapter.

2. The paper's description mainly focuses on solving the task of medical image annotation, yet it also includes experiments on natural data. I believe the use of video data does not align well with the primarily medical theme described in the paper.

3. Even if the method is shown to be extendable to natural data, using only one natural dataset is insufficient. To enhance its applicability and impact across broader domains within the NeurIPS community, I suggest using more varied types of data (e.g., natural images, video sequences).

4. Writing details. There are inconsistencies in figure captions, with some having periods and others not. The numbers in Table 1 exceed the table lines. Some tables comparing with other methods lack proper citations. The colors in the figures are too varied; please use black, white, and gray tones to highlight professionalism.

5. The explanation of the annotated data volume for weak supervision and full supervision is unclear.

6. Although the text mentions experiments conducted under low annotation budgets (2%-5%) and fully annotated scenarios, comparing the performance of different methods, it does not clearly explain why a 2%-5% annotation data volume is used in the fully annotated scenarios.

7. In practical experimental design, the reason for using a 2%-5% annotation data volume under full supervision is usually to compare the performance of weak supervision and full supervision methods on the same data volume basis. However, weak supervision and full supervision are different concepts, and using a partial annotation data volume for both does not align with the concept of full supervision.

**Questions:**

1. The main point of the authors' discussion is the time-consuming and labor-intensive nature of medical annotation. However, they added a dataset in a natural scenario, which seems somewhat off-topic. Is there a reason for this?

2. Why are fully supervised and semi-supervised methods compared using different annotation increments? Fully supervised data is entirely annotated. How does this relate to the primary focus of the paper on semi-supervised scenarios?

3. The paper introduces different loss functions, but what are the weight values for these different loss functions under different experimental settings? The paper does not clarify this point.

4. The ACDC dataset includes both weakly supervised and fully supervised experiments. Why does the MS-CMR dataset only have weakly supervised experiments, while the CHAOS and DAVIS datasets have fully supervised experiments?

**Limitations:**

1. Annotation costs and efficiency: Although the paper discusses the potential of active learning to reduce annotation costs, it would be beneficial to explicitly discuss the trade-offs between annotation quality and the computational expense of implementing the proposed methodology. This discussion could provide insights into practical considerations for real-world deployment.

2. Section 4.5 ablation study focuses solely on the combination of loss functions. Consider other ablation experiments, such as the scribble-weighted distance.

3. Due to limited computational resources, the researchers selected a single architecture for each type of annotation and limited the number of experimental runs and model training. This may result in an incomplete evaluation of other architectures and models' performance.

4. Video segmentation models typically require extensive pre-training to achieve good metrics. However, due to resource constraints, the researchers did not conduct such experiments, potentially leading to suboptimal performance in video segmentation tasks.

5. The paper does not adequately address the issue of training set bias, which may result in unfair outcomes for minority groups. Future work should consider this issue more comprehensively and focus on the method's applicability across different domains.

6. The proposed method mainly targets 3D segmentation tasks in the medical field. Further validation and adjustments may be necessary for applications in other fields.

7. Was the selection of annotated data random? Did it consider data distribution bias? This could result in unfair outcomes for minority groups.

---

> ### Author Rebuttal · Authors · 2024-08-07
>
> We thank the reviewer for their great feedback. We hope to address the main concerns for the reviewer here.
>
> ***Model performance and annotation time trade-offs for weakly and fully-supervised 2D slices and 3D volumes***
>
> Thank you for your feedback related to this. As stated in the general response, in Figure 5 of the attached PDF we have provided a graph that describes the relationship between model performance and annotation time for our method utilizing weakly and fully-supervised 2D slices and random sampling of fully-supervised 3D volumes on the ACDC dataset. This will help ensure a more fair comparison between the different active learning methods.
>
> ***Performance on Video Segmentation Dataset***
>
> Thank you for your feedback related to this. As stated in the general response, we additionally evaluated a segmentation model with a ResNet-50 backbone [1] pre-trained on ImageNetV2 on the video segmentation task and generated results which can be seen in Table 8 in the attached PDF. We can see that there is significant improvement on the scores for the video segmentation dataset, which is much more comparable to current methods.
>
> While the focus of our work is 3D medical segmentation, we wanted to demonstrate that our approach can be extended to other types of datasets. We picked video segmentation because there are several frames in a video, which is similar to having several slices in a volume. In future work, we will evaluate our model on additional video segmentation datasets.
>
> ***Writing Quality***
>
> Thank you for your feedback related to this. We will make sure to improve on this when revising.
>
> ***Additional questions***
>
> 1. As mentioned before, we wanted to demonstrate that our approach can be extended to other types of datasets. We picked video segmentation because there are several frames in a video, which is similar to having several slices in a volume.
> 2. Fully supervised and weakly-supervised methods are compared using the same proportion of data annotated. We have also extrapolated the performance vs. annotation time of the different methods in Figure 5 in the attached PDF. Please let us know if we misunderstood your question.
> 3. We have provided information about the loss weights in the general response. Please let us know if you would like any additional information.
> 4. Unfortunately, the fully supervised labels are unavailable for the MS-CMR training set. We have reached out to the dataset owners and have not received a reply. There are no manually generated scribbles for the CHAOS and DAVIS datasets. We have explored several automated methods to generate scribbles from the full labels, including skeletonization and random walk methods as specified in prior work [2]. However, we were unable to generate reasonable segmentation results with the automated methods.
>
> ***Discussion on limitations***
>
> 1. In Figure 5 in the attached PDF, we have provided performance and annotation time trade-offs for weakly and fully-supervised slice-based active learning and fully-supervised volume-based active learning
> 2. The last line of the ablation study includes the impact of scribble-weighted distance. The title “Scribble” may have been misleading and will be fixed in the camera-ready version.
> 3. We have provided video segmentation results using a stronger pretrained model in Table 8 of the attached PDF and the results are much more comparable to other video segmentation methods.
> 4. Unfortunately, we did not have the opportunity to consider training set bias which may lead to unfair outcomes to minority groups. The demographic data was unavailable for us to consider this. In future work, we will incorporate datasets which have this information to better address this issue.
> 5. We agree with this. In future work, we will consider evaluation on more varied datasets.
> 6. We agree that this could be an issue. Related to #4, in future work we will utilize datasets with demographic data to ensure that we are able to address this issue.
>
> Thank you again for reading our paper and your help and feedback!
>
> [1] MH Nguyen, D., Nguyen, H., Diep, N., Pham, T. N., Cao, T., Nguyen, B., ... & Niepert, M. (2024). Lvm-med: Learning large-scale self-supervised vision models for medical imaging via second-order graph matching. Advances in Neural Information Processing Systems, 36.
>
> [2] Valvano, G., Leo, A., & Tsaftaris, S. A. (2021). Learning to segment from scribbles using multi-scale adversarial attention gates. IEEE Transactions on Medical Imaging, 40(8), 1990-2001.

---

> ### Author Response · Authors · 2024-08-12
> **Additional Questions**
>
> Hope this comment finds you well. We just wanted to see if our rebuttal addressed your concerns and if you had any other questions about our paper. As there is only one day left in the discussion period, we wanted to make sure we have time to address any additional questions or concerns you may have.

---

### Official Review · Reviewer_ff5T · 2024-07-12

**Soundness:** 2
**Presentation:** 3
**Contribution:** 2
**Rating:** 5
**Confidence:** 4

**Summary:**

The authors propose a new metric learning approach to improve the Coreset method (an optimization framework to seek the most diverse samples for model learning in active learning) with applications for slice-based active learning in 3D medical segmentation. The main idea is to utilize data groups extracted from 2D slides in 3D volumes and build group-based contrastive learning to enhance feature representations. The authors conduct experiments on four datasets and compare them with several baselines in active learning. The experiment shows good overall performance.

**Strengths:**

One of the main strengths of this paper is that the authors compared diverse baselines and benchmarks with some of the latest methods, for e.g., [19,37]. Also, their method achieves good performance with significant improvements on four datasets. The author also provides ablation studies to analyze the contributions of key components in Table 3.

**Weaknesses:**

There are two main critical concerns of Reviewers:

(a) The way the author improves the Coreset with deep metric learning using group-based contrastive learning is limited in its novelty. In particular, group-based contrastive learning has already been introduced in prior work [65], and several advanced versions have been introduced. Although the authors modify with two sums in Equation (4), this is a minor point.  In short, given the claim that deep metric learning is the core contribution, authors should further tailor contrastive learning for 3D medical tasks, for e.g., making it robust under domain shift or considering other factors like hierarchical relations [1,2]. Also, fusing different metric learning using contrastive or generative learning is also a promising direction.

(b) Although experiments are promising, they are tested only with a single architecture, which is not enough to conclude whether the method is generalized to different architectures. As can be seen in Table 2 (DICE scores for DAVIS), model accuracy is still under an acceptable threshold; this suggests that further evaluation of current foundation models [3] (using SAM) or advanced pre-trained medical models [4] (has versions for both ResNet and ViT) is necessary to boost results further. In summary, it's important to provide further experiments with another architecture (preferred large-pre-trained models) to validate the robustness of the proposed active learning. Besides 3D volumes, considering one or two test cases with video is also interesting, where the group of similar 2D slices becomes a group of similar frames.

[1] Taleb, Aiham, et al. "3d self-supervised methods for medical imaging", NeurIPS 2020

[2] Zheng, Hao, et al. "Hierarchical self-supervised learning for medical image segmentation based on multi-domain data aggregation" MICCAI 2021

[3] Ma, Jun, et al. "Segment anything in medical images." Nature Communications 2024

[4] MH Nguyen, Duy, et al. "Lvm-med: Learning large-scale self-supervised vision models for medical imaging via second-order graph matching", NeurIPS 2023

[65] Sohn, Kihyuk. "Improved deep metric learning with multi-class n-pair loss objective." NeurIPS 2016

**Questions:**

My questions include:

(a) can the author explain the 'group' definition in Section 3.1? It is confusing whether this 'group' refers to a set of similar 2D slides inside a 3D volume of one patient or it refers to a set of 2D slides of different patients. Reviews did not understand what the author wrote in lines 123-128.

(b) how the algorithm 1 be extended if each patient has different data modalities like CT & MRI?

**Limitations:**

As mentioned in the weakness section, this paper needs to make a major improvement by either adding more experiments (with another architecture) or further improving the novelty in the way it formulates deep metric learning with contrastive ideas. The writing also need to be improved, especially when explaining the 'group' concept to avoid confusing readers.

---

> ### Author Rebuttal · Authors · 2024-08-07
>
> We thank the reviewer for their great feedback. We hope to address the main concerns for the reviewer here.
>
> ***Method novelty***
>
> Thank you for your feedback related to this. We addressed your concerns related to the method novelty in the general response and here further elaborate our novelty compared to the three suggested references. While group based contrastive learning has been proposed in the literature, as acknowledged in our paper [1], our overall framework with Coreset and contrastive learning and the application to the medical domain is very novel.
>
> To our knowledge, no other method has a comparable approach which is tailored to the medical domain. It is also easy to extend our approach to different groupings. In the paper, we have also provided an intuition on what groupings will generate useful features (i.e. groupings with larger intra-group differences are more likely to be useful). We also have provided our batch sampling technique, which is a nontrivial addition to our methodology because standard random data loading would yield minimal impact from our loss due to the low probability of randomly selecting two slices from the same group.
>
> Specifically, our group-based contrastive loss is very different from the multi-class n-pair loss [1]. We use explicit medical groupings in the data to generate positive and negative samples for a particular batch. While the other approach combines samples from different classes, we define hierarchical groups in an unsupervised setting and adapt the loss to address batch samples in the same hierarchical group. Generally in the loss slices that are not part of the same group would be negative samples. However, if there is a hierarchical relationship between slices, slices that are not part of the same subgroup but are part of the same more general group are ignored when computing the batch loss. This is seen in the denominator of Equation 4, which excludes patient slices for a particular data point that are not part of the same group. As mentioned in the paper, excluding non-group patient slices ensures that the model does not promote dissimilarity between non-group slices from the same patient. This ensures that we can sum multiple group losses together without counteracting their effects.
>
> While the other referenced hierarchical method also utilizes groupings in medical data [3], their groupings are assumed to be quite large and it’s unclear how they would extend their group classification approach to when there are a large number of patients, volumes, and adjacent slice groups. Our contrastive-based approach and our batch sampling technique addresses these problems and can also be extended to incorporate larger groups as well.
>
> In the other referenced method [4], while their approach is interesting they do not explicitly utilize any medical groupings in their method and it is not clear whether their method could be easily extended to incorporate that information as well.
>
> We will discuss these references and differences from our method in the final camera-ready version.
>
> ***Empirical evaluation using large pre-trained segmentation model***
>
> Thank you for your feedback related to this. As discussed in the general response, we have additionally evaluated our methodology with your suggested methodology using a ResNet-50 backbone [2]. For the medical tasks, we utilize a backbone pre-trained on medical images [2] and for the video segmentation task we utilize a backbone pre-trained on ImageNetV2. The results can be seen in Table 8 and improved the overall segmentation model performance.
>
> ***Additional questions***
>
> 1. For our proposed method and the paper implementation, the definition of “group” refers to a set of 2D slices associated with one patient. In some cases, this group can also be associated with one volume: for example, for the volume or slice-adjacency group. However, this may not be the case for the patient group, in which it is possible for slices from different volumes to be part of the same group if they are associated with the same patient. However, the definition of group can be extended to include 2D slices associated with multiple patients if a more general hierarchical group is used. One example is a pathology group which may contain slices from different patients.
> 2. Unfortunately, we did not have time to thoroughly consider utilizing multiple modalities like CT and MRI. However, it is an interesting question. Assuming we used the same segmentation model for both modalities, preliminarily, it seems like we would add an intermediate hierarchical level for modality in between volume (the more specific group) and patient (the more general group). We will look into this in future work.
>
> Thank you again for reading our paper and your help and feedback!
>
> [1] Sohn, K. (2016). Improved deep metric learning with multi-class n-pair loss objective. Advances in neural information processing systems, 29.
>
> [2] MH Nguyen, D., Nguyen, H., Diep, N., Pham, T. N., Cao, T., Nguyen, B., ... & Niepert, M. (2024). Lvm-med: Learning large-scale self-supervised vision models for medical imaging via second-order graph matching. Advances in Neural Information Processing Systems, 36.
>
> [3] Zheng, H., Han, J., Wang, H., Yang, L., Zhao, Z., Wang, C., & Chen, D. Z. (2021). Hierarchical self-supervised learning for medical image segmentation based on multi-domain data aggregation. In Medical Image Computing and Computer Assisted Intervention–MICCAI 2021: 24th International Conference, Strasbourg, France, September 27–October 1, 2021, Proceedings, Part I 24 (pp. 622-632). Springer International Publishing.
>
> [4] Taleb, A., Loetzsch, W., Danz, N., Severin, J., Gaertner, T., Bergner, B., & Lippert, C. (2020). 3d self-supervised methods for medical imaging. Advances in neural information processing systems, 33, 18158-18172.

---

> > ### Comment · Reviewer_ff5T · 2024-08-12
> > **Thanks for your rebuttal**
> >
> > Dear Authors,
> >
> > I thank you for your efforts to do additional experiments, for e.g., with pre-trained models as well as the explaining novelities in your contrastive algorithm. **I decided to increase my score from 4 -> 5**. If your paper is accepted, I would recommend:
> >
> > (a) Try to remove empty space in Figure 1 (then put it on the left side) and spend the right part to draw deeper details about your contrastive supervised learning tailored for 3D volumes. It's essential to highlight your contributions compared to other contrastive baselines.
> > (b) Write a good definition of the 'group' before you dive into the details of the proposed algorithm.
> > (c) Include new experiments with pre-trained models + other ablation studies asked by other reviewers.
> >
> > Regards

---

> > > ### Author Response · Authors · 2024-08-12
> > >
> > > Thank you! If our paper is accepted, we will make sure to incorporate these changes in the camera-ready version.

---

> ### Author Response · Authors · 2024-08-12
> **Additional Questions**
>
> Hope this comment finds you well. We just wanted to see if our rebuttal addressed your concerns and if you had any other questions about our paper. As there is only one day left in the discussion period, we wanted to make sure we have time to address any additional questions or concerns you may have.

---

### Official Review · Reviewer_gm1Z · 2024-07-13

**Soundness:** 3
**Presentation:** 3
**Contribution:** 3
**Rating:** 6
**Confidence:** 4

**Summary:**

The paper presents an active learning framework based on deep metric learning for 3D medical image segmentation. The proposed approach extends the CoreSet algorithm for active learning by computing distances on features learned via a contrastive loss. This contrastive loss exploits low-cost labels to group samples (2D slices) based on their spatial position and corresponding subject. Additionally, a weakly-supervised version of the method, based on scribbles, is presented. In this version, scribbles are used in an auxiliary distance term measuring the intensity difference for labelled pixels. Experiments are conducted on three medical imaging dataset and one video segmentation dataset to evaluate the method's performance. Results shows the method to outperform recent active learning approaches and baselines in the majority of test cases.

**Strengths:**

* The paper introduces a simple yet efficient way to incorporate contrastive learning in the CoreSet algorithm for active learning in segmentation.

* The extension of active learning to a weakly-supervised setting is also interesting and could have useful applications.

* Experiments are comprehensive. The proposed method is tested on four datasets and compared against several related approaches. Results show clear improvements in most cases.

* The paper is well written, and the method can be understood easily.

**Weaknesses:**

* Despite its novelty, the board idea of the proposed method is similar to previous work using contrastive learning for core set selection, for instance:

Ju J, Jung H, Oh Y, Kim J. Extending contrastive learning to unsupervised coreset selection. IEEE Access. 2022 Jan 13;10:7704-15.

Jin Q, Yuan M, Qiao Q, Song Z. One-shot active learning for image segmentation via contrastive learning and diversity-based sampling. Knowledge-Based Systems. 2022 Apr 6;241:108278.

While these previous approach do not use group labels as the proposed method, a similar strategy was employed for self-supervised segmentation of medical images:

Chaitanya K, Erdil E, Karani N, Konukoglu E. Contrastive learning of global and local features for medical image segmentation with limited annotations. Advances in neural information processing systems. 2020;33:12546-58.

* The comparison against other approaches might be unfair since the contrastive (pre) training, which is used to select the core set, also boosts the performance of the segmentation model (regardless of which labelled samples are used for training). This can be seen in the work of Chaitanya et al (see reference above), where a segmentation trained with the same contrastive loss achieves a performance near full supervision with very few samples.

* The scribbled weighted distance presented in Section 3.2 lacks motivation. It is unclear to me how the pixel-wise difference in intensity gives useful information for non-registered images (potentially of different subjects).

**Questions:**

* Can authors clarify the contribution of their work with respect to previous work on active learning based on a contrastive technique?

 * What is the meaning/definition of h_{S0} in Eq (3) ?

* Do you assume that pixels x^i_1 and x^i_2 correspond to the same anatomical structure in the scribble-weighted distance (i.e., images are registered)? Why compute the distance on pixel intensities?

* Why is the method worse than random selection on the CHAOS dataset (2% fully-supervised) ?

* Can you compare your method against random/maxEntropy selection on a segmentation model pre-trained using the SAME contrastive loss as your model?

**Limitations:**

* The last section of the paper mentions relevant limitations of the work.

---

> ### Author Rebuttal · Authors · 2024-08-07
>
> We thank the reviewer for their great feedback. We hope to address the main concerns for the reviewer here.
>
> ***Method novelty***
>
> Thank you for your feedback related to this. We have addressed several of your concerns related to method novelty in the general response when discussing how our approach, unlike others [1-2], is uniquely tailored to the 3D medical segmentation task. We note that the focus of the other contrastive learning work in 3D medical segmentation [3] is self-supervised pre-training. Thus, the generated contrastive features may not be optimal for the Coreset-based active learning objective, as shown in the ablation results (Table 3 of our paper) when comparing the generated contrastive features from a standard contrastive loss and our proposed loss. While they utilize local and global features, they do not utilize domain information to generate the groupings which we have demonstrated is beneficial for the active learning task. Additionally, self-supervised pre-training is outside the scope of this work and we do not dispute that it is likely there may be additional performance benefits with self-supervised pre-training. We also compare our method to TypiClust [4], another contrastive learning-based active learning method, and demonstrate that our medically-tailored approach produces better results (Table 1 of our paper). We will discuss these references and differences from our method in the final camera-ready version.
>
> ***No self-supervised pre-training for fair comparison***
>
> Thank you for your feedback related to this. In order to ensure a fair comparison between all the active learning methods, the segmentation model does not utilize the pre-trained contrastive encoder. There are some practical engineering benefits if the active learning and segmentation models are decoupled. We hypothesize that there may be more performance improvements if the pre-trained contrastive encoder is utilized by the segmentation model.
>
> ***Scribble-weighted distance***
>
> Thank you for your feedback related to this. The ACDC challenge data is roughly registered and others have utilized the correspondence between the local image regions for self-supervised pre-training on the ACDC dataset [3]. We compared several different approaches of computing the scribble-weighted features (including segmentation model features as well as segmentation model predictions) and found that the best performance was obtained with the pre-processed pixel intensities.
>
> ***Additional questions***
>
> 1. We hopefully clarified the contribution of our work related to similar work. Please let us know if there are any additional questions.
> 2. h_{S0} in Eq (3) refers to the scribble-weighted distance. The {S0} part is technically unnecessary but we wanted to emphasize that the scribble-weighted distance is calculated based on the annotations that are currently available.
> 3. As mentioned prior, we do assume that the images are roughly registered and we compare the pre-processed pixel intensities because it obtained the best performance during the validation testing. This may be because of the anatomical region correspondence between images.
> 4. Our method does perform worse than the random selection on the CHAOS dataset for the 2% fully-supervised data point. We want to emphasize that random selection is still considered an extremely competitive baseline in active learning [5]. However, in our paper, we demonstrate that our method achieves the best performance on 19 out of 24 comparison points in comparison to the diversity-based methods (Coreset, VAAL, TypiClust, Random) and 22 out of 24 comparison points in comparison to the entropy-based methods  (BALD, Variance Ratio, Stochastic Batches) from Table 1 and 2 in our paper.
> 5. As mentioned prior, we do not perform self-supervised pre-training for fair comparison between the methods.
>
> Thank you again for reading our paper and your help and feedback!
>
> [1] Ju, J., Jung, H., Oh, Y., & Kim, J. (2022). Extending contrastive learning to unsupervised coreset selection. IEEE Access, 10, 7704-7715.
>
> [2] Jin, Q., Yuan, M., Qiao, Q., & Song, Z. (2022). One-shot active learning for image segmentation via contrastive learning and diversity-based sampling. Knowledge-Based Systems, 241, 108278.
>
> [3] Chaitanya, K., Erdil, E., Karani, N., & Konukoglu, E. (2020). Contrastive learning of global and local features for medical image segmentation with limited annotations. Advances in neural information processing systems, 33, 12546-12558.
>
> [4] Hacohen, G., Dekel, A., & Weinshall, D. (2022). Active learning on a budget: Opposite strategies suit high and low budgets. arXiv preprint arXiv:2202.02794.
>
> [5] Munjal, P., Hayat, N., Hayat, M., Sourati, J., & Khan, S. (2022). Towards robust and reproducible active learning using neural networks. In Proceedings of the IEEE/CVF Conference on Computer Vision and Pattern Recognition (pp. 223-232).

---

> > ### Comment · Reviewer_gm1Z · 2024-08-08
> > **More clarifications needed**
> >
> > Thank you for the clear and detailed answers. I understand that [3] uses its group-wise contrastive loss for self-supervised pretraining, and not specifically for active learning. However, one can also argue that your main contribution is applying the well-known CoreSet method on the representation learned by this existing contrastive loss.
> >
> > Moreover, I still do not fully understand why the contrastive learning-based encoder is not used in the downstream segmentation task. Since active learning methods only differ in how they select the samples to annotate, the comparison would be fair if all methods use the same training strategy with their own selected annotations. For example, the random selection strategy could simply fine-tune the contrastive learning-based encoder + the decoder using annotations of randomly selected images.
> >
> > Last, I still have some doubts regarding the alignment of structures in the scribble-weighted distance. In ACDC, even though images are globally registered, the position and size of cardiac structures rarely match across different images due to the movement of the heart and respiration. The local contrastive loss in [3] compares local regions in the *same* image. From my understanding, the images in your eq (5) can be different ?

---

> > > ### Author Response · Authors · 2024-08-09
> > > **Additional Clarification**
> > >
> > > Thank you for your additional comments. We can respond to them here.
> > > > However, one can also argue that your main contribution is applying the well-known CoreSet method on the representation learned by this existing contrastive loss.
> > >
> > > Our loss has some important differences with the global contrastive loss in the other method.  The big difference is that we have a framework for including different types of groupings and also combining the different losses. The global contrastive loss has only one type of grouping: sequential volume partitions. We group by patient, volume, adjacent slice groups, and provide guidance on incorporating more general groups. These groupings are hierarchical and when calculating the batch loss utilizing all three groupings, we would need to mask samples that are not part of the specific group but are in more general hierarchical groups so we do not unintentionally counter the effectiveness of one of the other losses. The effectiveness of combining these hierarchical groups in our loss is demonstrated in our ablation results (Table 3 of our paper) . The other method does not address how best to combine different hierarchical groups or groupings in general and does not have to worry about such issues. Additionally, because of the careful design of our loss formulation and our batch sampling method, we are able to train our loss in one stage whereas the other method requires a two-stage training process. Our results tailored to the 3D medical segmentation application use case can be very useful to other researchers working in this space.
> > >
> > > > Moreover, I still do not fully understand why the contrastive learning-based encoder is not used in the downstream segmentation task.
> > >
> > > One of the main issues is architecture incompatibility: the contrastive encoder may not be readily used by, for example, our weakly-supervised segmentation architecture and we would have to adapt the architecture, which we felt would be out of the scope of our current work. Thus, in order to ensure a fair comparison between all methods, none of the downstream segmentation models used the contrastive learning-based encoder. However, especially for the fully-supervised networks, we can definitely generate some results with the contrastive learning-based encoder and include this in our camera-ready version.
> > >
> > > > Last, I still have some doubts regarding the alignment of structures in the scribble-weighted distance…The local contrastive loss in [3] compares local regions in the same image. From my understanding, the images in your eq (5) can be different ?
> > >
> > > Yes, the images in equation 5 can be different in our method which seems to be similar to the global contrastive loss in [3]. We can register the images and generate some updated results in our camera-ready version of the paper. While validation testing revealed a small performance boost with the existing method, further aligning the structures may produce better results.

---

> > > ### Author Response · Authors · 2024-08-12
> > > **Additional Experiment Results**
> > >
> > > We have produced some additional experiment results to address some of your concerns.
> > >
> > > First, we considered utilizing the contrastive learning-based encoder in the downstream segmentation task. We utilized the fully-supervised segmentation architecture that we referenced in the general rebuttal [6] and we compared the results with utilizing random weight initialization, pre-trained medical weights provided by [6], and our original U-Net trained from scratch. We generated the results on a fully-supervised segmentation task on the ACDC dataset and calculated the mean DICE scores over the 2%-5% annotation points (for fair comparison). The results are as follows:
> > >
> > > |  | **UNet** | **LVM (not pre-trained)**  | **LVM (contrastively pre-trained with GCL)** | **LVM (medically pre-trained)** |
> > > |--------------|--------------|--------------|--------------|--------------|
> > > | **Random**    |  75.7 | 45.1 | 65.9 | 82.6 |
> > > | **Stochastic**    | 69.9 | 47.4 | 67.7 | 82.9 |
> > > | **Coreset**    | 71.2 | 48.0 | 68.5 | 81.8  |
> > > | **Ours**    | \*78.1\*  | \*52.2\*| \*69.3\* | \*83.6\*  |
> > >
> > > We note that the segmentation results that leveraged the contrastive weights were better than training from scratch but worse than using the medical pre-trained weights or the U-Net. This is in line with our general observation that weights useful to the downstream task may not necessarily correlate to weights that are useful for the coreset objective and vice-versa.
> > >
> > > Second, in order to consider improving the scribble-weighted distance, we considered improving the alignment of structures in the volumes by performing group-wise registration using symmetric normalization based on each cardiac phase. After re-sampling and re-sizing all the volumes to 128 x 128 x 32, we used the average volume for each phase as the template for group-wise registration. After registering the volumes and the scribble masks, when computing the scribble-weighted distance between slices, we utilized the registered slice and mask instead. We considered a variety of weights for the scribble-weighted distance function and ultimately found that there was no improvement in using this approach.
> > >
> > > We will make sure to include these results in the final camera-ready version. Please let us know if you have any additional questions or comments.
> > >
> > > [6] MH Nguyen, D., Nguyen, H., Diep, N., Pham, T. N., Cao, T., Nguyen, B., ... & Niepert, M. (2024). Lvm-med: Learning large-scale self-supervised vision models for medical imaging via second-order graph matching. Advances in Neural Information Processing Systems, 36.

---

> > > > ### Comment · Reviewer_gm1Z · 2024-08-12
> > > > **Thanks for the answers**
> > > >
> > > > I am satisfied with the authors' detailed answers to my comments, and wish to upgrade my score to Weak Accept.

---

> > > > > ### Author Response · Authors · 2024-08-12
> > > > >
> > > > > Thank you!

---

### Author Rebuttal · Authors · 2024-08-07

Thank you all for reading our paper and your great feedback. I will address some of your general questions in the following response. Please note that we have attached a PDF with an additional table and figure that we reference in our response.

***Method novelty***

We want to emphasize the unique application area of our proposed method, which is slice-based active learning for 3D medical segmentation. Our proposed method is specifically tailored to utilizing individual slices for 3D medical segmentation, which is more cost-effective than using full volumes. We propose a unique Group Contrastive Learning (GCL) framework which leverages patient, volume, and adjacent slice relationships which are groupings inherent to medical imaging. Furthermore, we provide ablation results which emphasize the benefit of these groupings for active learning in 3D medical segmentation and also provide intuition on what groupings will generate useful features (i.e. groupings with larger intra-group differences are more likely to be useful). Additionally, in Figure 5 in our attached PDF we provide a graph which demonstrates the cost-effectiveness of our approach in comparison to volume-based active learning.

The other Coreset-based methods which use contrastive features [1-2] are not tailored to efficiently learning 3D medical segmentation models. For example, the volume group in our method specifically reduces the number of redundant slices annotated from similar volumes.  The other approaches do not specifically focus on reducing slice annotations per volume. Additionally, in our ablation study, we demonstrate that the Coreset algorithm with contrastive features generated from our proposed method produces much better performance than the Coreset algorithm with standard contrastive features not tailored to the 3D medical segmentation task.

***Empirical evaluation using large pre-trained segmentation model***

We additionally evaluated our method and the top three comparison methods using a large segmentation model with a ResNet-50 backbone [3], pretrained on medical images [3] and ImageNetV2. We noted that there was a performance improvement for the fully-supervised task and strong performance even with 1% of the data. Results are in Table 8 in the attached PDF. We do not report weakly-supervised results because the weakly-supervised architectures cannot easily utilize pre-trained backbones.

***Model performance vs. annotation time for different methods***

As some reviewers suggested, in Figure 5 in the attached PDF we provided a graph which describes the relationship between model performance and annotation time for our method utilizing weakly and fully-supervised 2D slices and random sampling of fully-supervised 3D volumes on the ACDC dataset. We follow prior work which assumes that annotators can annotate scribbles 15x as fast as the full masks [4]. To ensure a fair comparison between the different methods, we do not incorporate any of the results from the pre-trained segmentation models. For the 3D results, similar to the 2D U-Net we train a 3D U-Net from scratch. Given comparable annotation time, our methods trained on both weakly and fully-supervised 2D slices far exceed the performance of random sampling of 3D volumes and achieve 3D volume maximum performance (with the given budget) with much less annotation time.

***Loss Weights***

In our experiments and ablation study, a 1.0 weight was applied whenever the NT_Xent loss was used. Additionally, a 1.0 weight was applied if only one group contrastive loss was used. In the ablation study on the ACDC dataset, for the experiments with multiple group contrastive losses, we tried different combinations of weights for the group contrastive losses and reported the best results. We will utilize the formulation in Appendix A for the loss weights and report them as a four-tuple (a,b,c,d) which corresponds to ($\lambda_1$,$\lambda_2$,$\lambda_3$,$\lambda_4$). A weight of 0 indicates the loss was not used. The combinations we tried are as follow (with the best bolded):

- patient and volume group loss without NT_Xent loss: (0, 0.125, 0.875, 0), **(0, 0.50, 0.50, 0)**
- volume and slice group loss without NT_Xent loss: (0, 0, 0.125, 0.875), **(0, 0, 0.50, 0.50)**
- patient and volume group loss with NT_Xent loss: (1,0. 10 , 0.35, 0), (1, 0.117, 0.233, 0), **(1, 0.05, 0.35, 0)**
- patient, volume, and slice group loss without NT_Xent loss: (0, 0.05, 0.25, 0.7), **(0, 0.33, 0.33, 0.33)**
- patient, volume, and slice group loss with NT_Xent loss: (1, 0.10, 0.20, 0.05), **(1, 0.05, 0.35, 0.025)**

We utilized the best loss/weight combination for our ACDC experiments. For the CHAOS, MSCMR, and DAVIS datasets, because there is only one volume per patient and thus no difference between the patient and volume loss, we tested two loss/weight combinations:
- volume loss with weight 0.35 with NT_Xent loss
- volume loss with weight 0.10 and slice loss with weight 0.30 with NT_Xent loss

Both weight combinations performed better than other comparison methods though volume loss with weight 0.35 with NT_Xent loss performed slightly better than the other combination.

[1] Ju, J., Jung, H., Oh, Y., & Kim, J. (2022). Extending contrastive learning to unsupervised coreset selection. IEEE Access, 10, 7704-7715.

[2] Jin, Q., Yuan, M., Qiao, Q., & Song, Z. (2022). One-shot active learning for image segmentation via contrastive learning and diversity-based sampling. Knowledge-Based Systems, 241, 108278.

[3] MH Nguyen, D., Nguyen, H., Diep, N., Pham, T. N., Cao, T., Nguyen, B., ... & Niepert, M. (2024). Lvm-med: Learning large-scale self-supervised vision models for medical imaging via second-order graph matching. Advances in Neural Information Processing Systems, 36.

[4] Valvano, G., Leo, A., & Tsaftaris, S. A. (2021). Learning to segment from scribbles using multi-scale adversarial attention gates. IEEE Transactions on Medical Imaging, 40(8), 1990-2001.

---

### Decision · Program_Chairs · 2024-09-25

**Decision:**

Accept (poster)

**Comment:**

While this work initially received mixed scores, authors addressed positively most of the reviewers comments, which led to an overall increase of the recommendation scores for this submission. I strongly recommend the authors to consider the reviewer comments in the camera-ready version, and include requested changes, such as i) deeper details about the contrastive supervised learning tailored for 3D volumes (Fig 1) to highlight the contributions this work, ii) experiments with pre-trained models, as well as other ablation studies used to address the raised concerns.